

# Size-Resolved Atmospheric Ice Nucleating Particles during East Asian Dust Events

Jingchuan Chen[1], Zhijun Wu[1,2*], Jie Chen[1], Naama Reicher[3], Xin Fang[1], Yinon Rudich[3], and Min Hu[1,2]

[1]State Key Joint Laboratory of Environmental Simulation and Pollution Control, College of Environmental Sciences and Engineering, Peking University, Beijing 100871, China
[2]Collaborative Innovation Center of Atmospheric Environment and Equipment Technology, Nanjing University of Information Science and Technology, Nanjing 210044, China
[3]Department of Earth and Planetary Sciences, Weizmann Institute of Science, 76100 Rehovot, Israel

*Correspondence to*: Zhijun Wu (zhijunwu@pku.edu.cn)

**Abstract.** Asian dust is an important source of atmospheric ice nucleating particles (INPs). However, the freezing activity of airborne Asian dust, especially its sensitivity to particle size, is poorly understood. In this study we report the first INP measurement of size-resolved airborne mineral dust collected during East Asian dust events. The measured total INP concentrations in the immersion mode ranged from $10^{-2}$ to $10^2$ $L^{-1}$ in dust events at temperatures between -25 and -5 ℃. The average contributions of heat-sensitive INPs at three temperatures, -10, -15, and -20 ℃, were $81 \pm 12\%$, $70 \pm 15\%$, and $38 \pm 21\%$, respectively, suggesting that proteinaceous biological materials have a substantial effect on the ice nucleation properties of Asian atmospheric mineral dust at warm temperatures. The dust particles which originated from China's northwest deserts are more efficient INPs compared to those from northern regions. There was no significant difference in the ice nucleation properties between East Asian dust particles and other regions in the world. An explicit size dependence of both INP concentration and surface ice active density was observed. The nucleation efficiency of dust particles increased with increasing particle size, while the INP concentration first increased rapidly and then levelled, due to the significant decrease in the number concentration of larger particles. A new set of parameterizations for INP activity based on size-resolved nucleation properties of Asian mineral dust particles were developed over an extended temperature range (-35 ~ -6 ℃). These size-dependent parameterizations require only particle size distributions as input, and can be easily applied in models.

## 1 Introduction

Ice formation in tropospheric clouds significantly impacts the microphysical processes and lifetime of clouds, thereby determining radiative forcing, precipitation, and the hydrological cycle (Lohmann and Feichter, 2005;Boucher et al., 2014;Lohmann et al., 2016). Mineral dust particles can act as ice nucleating particles (INPs) that trigger heterogeneous ice nucleation at relatively high temperatures and low relative humidities by efficiently lowering the energy barrier to form the critical ice embryo (Pruppacher and Klett, 1997). Up to 2000 Tg of mineral dust are emitted from arid and semiarid areas into



the troposphere annually, and their estimated atmospheric loading is the largest among different types of aerosol particles (Prospero, 1999;Textor et al., 2006).

Asian deserts, especially East Asian deserts, are among the major sources (~800 Tg per year) of mineral dust to the global troposphere (Zhang et al., 1997). Asian dust can be transported across the Pacific Ocean, sometimes completing more than 35 one full cycle around the globe (Sun et al., 2001;McKendry et al., 2008;Uno et al., 2009). Ground-based and aircraft in situ cloud measurements have confirmed that dust and adsorbed biological aerosols from regions such as Asia may affect ice formation in mid-level clouds (Creamean et al., 2013;Pratt et al., 2009). In addition, modelling studies suggest that Asian dust affects ice particles' formation in mixed-phase clouds (Wiacek et al., 2010). However, simplified parameterizations that illustrate INP activity are required to accurately predict the occurrence and evolution of clouds, suggesting that there is a need 40 for measurements of ice formation on different INP types (Gultepe et al., 2017;Korolev et al., 2017).

In recent decades, many studies attempted to quantify the efficiency of mineral dust particles based on field observations (DeMott et al., 2010;Boose et al., 2016a) and laboratory experiments, investigating worldwide surface-collected dust samples(Niemand et al., 2012;Kaufmann et al., 2016), airborne desert dust(Price et al., 2018;Schrod et al., 2017), reference minerals(Atkinson et al., 2013;Kanji et al., 2008;Knopf et al., 2020), and commercially available dust such as Arizona test 45 dust (Niemand et al., 2012;Kaufmann et al., 2016;Kanji et al., 2008). Dust particles are mainly composed of mineral components, including clay minerals (containing illite, kaolinite, chlorite, etc.), quartz, feldspar, calcite, and so on (Murray et al., 2012;Scanza et al., 2015;Tang et al., 2016). Because of the high content (Murray et al., 2012) and increasing ratio during long-range transport of atmospheric dust (Margaret et al., 1994;Uno et al., 2009), clay minerals are widely investigated in ice nucleation studies (Mason, 1960;Eastwood et al., 2008;Pinti et al., 2012;Wex et al., 2014). However, recent studies have 50 demonstrated that K-feldspars are the most efficient mineral INP (Atkinson et al., 2013;Zolles et al., 2015;Whale et al., 2017;Harrison et al., 2019;Yakobi-Hancock et al., 2013;Harrison et al., 2016;Peckhaus et al., 2016;Kiselev et al., 2017). As a major component in mineral dust, quartz is considered a common supplement to feldspars, which may explain the freezing observed at lower temperatures (Boose et al., 2016b;Reicher et al., 2019).

In addition to the active components in mineral dust particles, the effect of particle size on ice nucleation activity, although 55 less explored compared to mineralogical properties, has been investigated in several studies (DeMott et al., 2015;Reicher et al., 2019;Porter et al., 2020). Larger particles often nucleate ice at lower relative humidities (Archuleta et al., 2005) and higher temperatures (Lüönd et al., 2010). Studying different mineral species (such as montmorillonite, kaolinite, illite, and Arizona test dust) at different particle sizes (ranging from 0.1 to 5.6 μm) demonstrated that there is an explicit size dependence in heterogeneous freezing (Welti et al., 2009;Kanji and Abbatt, 2010;Ladino et al., 2011;Hoffmann et al., 2013;Reicher et al., 60 2019). Furthermore, biological materials can adhere to the surfaces of larger dust particles during atmospheric lifting and transport processes, and effectively enhance the ice nucleation ability at higher temperatures (Creamean et al., 2013;O'Sullivan et al., 2016;Maki et al., 2018).

Despite their importance, few ice nucleation investigations focused on collected airborne Asian dust particles, and their nucleation activities and particle properties (Bi et al., 2019;Jiang et al., 2016;Iwata and Matsuki, 2018). In fact, reference single





mineral dust and surface-collected particles do not fully represent the actual dust transport process in the troposphere, which includes gravitational dust sedimentation, adsorption of biological materials, and complex atmospheric chemical aging reactions. Moreover, parameterizations are mainly based on the active mineral components of dust, but the effect of particle size is usually not considered. Although the same mineral components are present over a wide range of particle sizes, their concentrations in different sizes can vary greatly. Margaret et al. (1994) have demonstrated that quartz and plagioclase exist

in significantly higher amounts in the 2 to 20 μm size fraction, while kaolinite is more common in the < 2 μm sizes. This variation affects the ice nucleation activity at different particle size fractions. Therefore, a quantitative investigation of size- and temperature-dependent ice nucleation properties of airborne Asian mineral dust is required for a better understanding of the role of Asian dust, especially after long-range transport, in influencing ice formation in clouds, and its consequences for climate change.

To further understand the ice nucleation efficiency and its sensitivity to Asian dust particle size, airborne mineral dust samples were collected during East Asian dust events in the spring of 2018 and 2019. The INP measurements were conducted by a cold-stage technique. We quantified the INP concentrations, surface active site densities, and the contribution of heat-sensitive INPs (mostly recognized as proteinaceous biological materials at warm temperatures) of mineral dust in different size classes. Furthermore, based on the measurements, a new set of parameterizations were developed, assuming that particle size class can

reflect the particle ice nucleation ability. Only particle size distributions are required as input in our parameterizations to predict ice nucleation activity when applied in models.

## 2 Materials and Methods

### 2.1 Sample Collection and Analysis

The sampling and measurements were performed at the Peking University Atmosphere Environment Monitoring Station

(PKUERS, 39.99°N, 116.31°E), an integrated observation laboratory in the northwestern urban area of Beijing. A weather station provided high time resolution meteorological parameters, including wind speed, wind direction, ambient relative humidity, and temperature.

Aerosol samples were collected on polycarbonate filters (47 mm Nuclepore, Track-Etch Membrane, 0.2 μm pores, Whatman) using the Micro-Orifice Uniform Deposit Impactor (MOUDI, MSP Corporation, USA). An 8-stage inertia impactor (Model

100-R) was used to collect size-resolved aerosol samples during dust events in the spring of 2018 and 2019. Size-resolved stages from 1 to 8 of the MOUDI with cut-points ($D_{50}$) ranging from 10 to 0.18 μm in aerodynamic diameters, at a flow rate of 30 L min$^{-1}$ were detected in this study (Marple et al., 1991). The sizes of collected particles in a given stage are set in a size bin, which is presented as the cut-point ($D_{50}$) in the following text. For example, most of the particles with an aerodynamic diameter less than 10 μm and larger than 5.6 μm will be collected on stage $D_{50} = 5.6\ \mu m$, therefore the size range between 5.6

and 10 μm will be presented here as cut-point $D_{50} = 5.6\ \mu m$. Fourteen sets of MOUDI samples were collected: 8 in 2018, and



6 in 2019. The sampling time varied with the duration of the dust events, which ranged from 5 to 24 hours. More details of the sampling are available in Table 1.

Particle bounce on the filters is the main cause of interstage losses, especially for larger particles. Due to the limitation of substrates and solid particles, the bounce is unavoidable. However, as many previous studies showed, the losses rapidly

decreased as the particles size decreased, and the collection efficiency curves of each stage are sharp, indicating that MOUDI is a high accuracy cascade impactor for collecting size-resolved particle samples, and was widely used in many laboratory experiments and field measurements (Marple et al., 1991;Mason et al., 2015;Mason et al., 2016;Si et al., 2018;Reicher et al., 2019).

## 2.2 INP Measurements

INP measurements in the immersion mode were conducted using the PeKing University Ice Nucleation Array (PKU-INA). PKU-INA is a cold stage-based ice nucleation array, operated after careful temperature calibration (Chen et al., 2018a). The sampled filter of each stage was completely submerged in 20 mL double-distilled water (resistivity of 18.2 MΩ·cm at 25 °C). An ultrasonic shaker was used to wash the particles off from the filters. The elution process lasted 30 minutes, and was kept in an ice-water bath to avoid the effects of ultrasound-induced heating. The ultrasonic shaker is effective for particle removal

into the water solution, and is widely used (Reicher et al., 2019;Chen et al., 2018b;Ardon-Dryer and Levin, 2014).

The particle suspension was immediately measured in PKU-INA. Briefly, 1 µL liquid droplets were pipetted onto a hydrophobic glass slide (40 mm in diameter, supplied by Marienfeld company, Germany) located on the cold stage (LTS120, Linkam, UK). As many as 90 droplets were pipetted dropwise in an experiment, and another glass slide was placed above to prevent the Wegener–Bergeron–Findeisen process (Jung et al., 2012). Droplets were cooled to 0 °C with a cooling rate of 10 °C

115   min[-1], because freezing will not be expected and observed in this temperature range, and then cooled at a rate of 1 °C min[-1] until all the droplets froze. A charge-coupled device (CCD) camera was used to record the measurements and monitor phase transitions based on the droplets' brightness, with a given time resolution of 1 frame per six seconds. To avoid frost during cooling, high-purity dry nitrogen was delivered into the cold stage during the measurement. The temperature uncertainty was less than ± 0.4 °C at the various cooling rates (Chen et al., 2018a).

The cumulative concentration of sites active above $T$ per unit sample volume is given by Vali (1971) and Vali et al. (2015):

$$K(T) = \frac{-\ln\left(1 - f_{ice}(T)\right)}{V} \text{ (cm}^{-3} \text{ of water)} \tag{1}$$

where $f_{ice}(T)$ is the fraction of droplets frozen above temperature $T$, and $V$ is the volume of each pipetted droplet (1 µL). Combined with the total sampling volume, the cumulative number concentration of INP ($N_{INP}$) per unit volume of sampled air is calculated in Eq. (2):

$$N_{INP}(T) = \frac{-\ln\left(1 - f_{ice}(T)\right)}{V_{air}} \text{ (L}^{-1} \text{ air)} \tag{2}$$





where $V_{air}$ is the total volume of sampled air per droplet converted to standard conditions (0 °C and 1013 hPa) during each particle collection period. To quantify and compare the ice nucleation activity of different size samples, the cumulative ice nucleation active site density $n_s$ (Connolly et al., 2009;Hoose and Mohler, 2012;Niemand et al., 2012), i.e., the number of active sites per unit surface area of INPs (Vali et al., 2015), is converted from the INP concentration and calculated using:

$$n_s(T) = \frac{N_{INP}(T)}{A} \; (\text{m}^{-2}) \tag{3}$$

where $A$ is the total surface area of the particles per unit volume of sampled air per droplet, based on the particulate matter information derived from size distribution measurements of the particles (see Section 2.3 below).

The uncertainty of the experimental results arises mainly from the representativeness of testing droplets for the total suspension. The population number of INPs present in the washing suspension is usually small, and the number of examined droplets is limited (90 droplets). Therefore, confidence intervals for the number of ice nucleating particles per unit sample volume (per droplet) should be considered. Using O'Sullivan's method (O'Sullivan et al., 2018;Barker, 2002), the confidence intervals are calculated in Eq. (4):

$$\mu(T) + \frac{(Z_{\alpha/2})^2}{2n} \pm Z_{\frac{\alpha}{2}} \left[ 4\mu + \frac{(Z_{\frac{\alpha}{2}})^2}{n} \right]^{0.5} /(4n)^{0.5} \tag{4}$$

where $\mu(T)$ is the number of INPs per droplet, $n$ is the droplet number, $Z_{\alpha/2}$ is the standard score at a confidence level $\alpha/2$, which is 1.96 for a 95% confidence interval.

Above mentioned parameters ($f_{ice}(T), K(T), N_{INP}(T), n_s(T)$, and uncertainty) are calculated for each size class. We sum size-resolved $N_{INP}(T)$ ranging from 0.18 to 10 μm during each dust event to get the total INP concentration. Gross $n_s(T)$ is directly derived from total $N_{INP}(T)$ by dividing the total $N_{INP}(T)$ by the total surface area of particles in all size classes.

## 2.3 Particulate Matter Measurements

Particle number-size distributions of 3 nm to 10 μm particles were measured during all dust events by a set of two scanning mobility particle sizers (SMPS, TSI Inc., USA), and an aerodynamic particle sizer (APS, Model 3321, TSI Inc., USA). The first SMPS, consisting of a short differential mobility analyzer (DMA, Model 3085) and an ultra-condensing particle counter (UCPC, Model 3776, with a flow rate of 1.5 L min⁻¹), was used to measure the 3 to 45 nm (stokes diameter) particles. The second SMPS with a long DMA (Model 3081) and a CPC (Model 3775, with a flow rate of 0.3 L min⁻¹) measured the 45 to 698 nm (stokes diameter) particles. The APS, with a flow rate of 1 L min⁻¹, was used to measure the 0.7 - 10.0 μm (aerodynamic diameter) particles. The time resolution for both SMPS and APS was 5 min.

Particle surface area distributions were derived from the number-size distributions assuming that the particles were spherical. The Stokes diameter measured by SMPS was transformed to aerodynamic diameter assuming a particle density of 1.5 g cm⁻³, which is commonly used for small size particles in urban atmospheres (Hu et al., 2012;Qiao et al., 2018). In order to estimate the total surface area sampled on each MOUDI stage, similar to the approach in Reicher et al. (2019), a transformation matrix





between the size distribution instruments and the MOUDI stages was applied, which was based on the particle cut-off characteristics of each stage and inter-stage particle losses reported in Marple et al. (1991).

Compared with the transformation matrix used by Reicher et al. (2019), the method adopted in this study has two optimizations.
Firstly, the size distribution instruments (SMPS and APS) have more sampling channels (64 channels). The more data points corresponding to each sampling particle size range the instruments measure, the higher accuracy for total surface area calculated. Secondly, according to the particle collection efficiency curves in Marple et al. (1991), narrowing the intervals of the transformation matrix better reflects the real distribution.

**2.4 Air mass back trajectory analysis**

Back trajectories of air mass arriving at the sampling site were calculated using the HYSPLIT 4 (Hybrid Single-Particle Lagrangian Integrated Trajectory) model of the NOAA Air Resources Laboratory (Stein et al., 2015). When computing archive trajectories, the GDAS (Global Data Assimilation System, 1 degree, global) was selected as meteorological input data. The 72-hour back trajectories were initiated at the beginning of each sampling period, and started a new trajectory every 1 or 2 hours until the end of the sampling period. MeteoInfoMap, a GIS application that enables the user to visualize and analyze the
spatial and meteorological data with multiple data formats (Wang, 2014) was used for final analysis and mapping.

**3 Results and Discussion**

**3.1 INP Concentrations**

Detailed sampling and measurement information for the 14 sets of samples collected in 2018 and 2019 are given in Table 1. The total sampled volumes and $PM_{10}$ mass concentrations during the sampling periods determine the particle collection
quantities, and affect the INP concentrations. Two weather condition scenarios, dust and non-dust events, were defined, based on $PM_{10}$ mass concentration (larger than 200 μg m$^{-3}$ lasting more than 2 hours for dust events), the volume concentration of coarse mode particles (mean concentration higher than 75 μm$^3$ cm$^{-3}$ for dust events; Wu et al., 2009), phenomenological dust storm observations operated by China Meteorological Administration (CMA, being reported as the largescale dust events), and the concentration of aluminium (Al) element (see Supplementary Information, referred to as SI from here on, Table S1).
Sample M4 was classified as a non-dust event. Sampling started at the end of a continuous dust storm period (M1, M2, and M3), and the air mass passed through the Bohai Sea before arriving in Beijing, therefore it was not dominated by mineral dust (see the SI, Fig. S1).

Results of all freezing curves containing airborne dust particles at various particle size classes are presented in pale yellow in Fig. 1. Each curve corresponds to one sampled filter. Freezing was observed from -5 to -25 °C for ambient filters, while blank
filters froze between -20 and -30 °C. The freezing temperature range of the blank filters is similar to that of distilled water, far lower than that of ambient samples, indicating a low contribution of contaminants from the filters. Frozen fraction curves from event M1 are also shown in Fig. 1, and each colour depicts a different size class ranging from 0.18 to 10.0 μm. Droplets



containing different size classes froze at different temperatures. Large particles froze at higher temperatures, while smaller particles froze at lower temperatures, indicating differentiated ice nucleating abilities. This will be discussed in more detail below.

The total concentrations of INPs ($N_{INP}$) as a function of temperature of all 14 samples, including 13 dust events and 1 non-dust condition (M4), are shown in Fig. 2 (a), marked by different colors. These results were obtained from the data in Fig. 1 using Eq. (2). In the sampling process, the eight-stage MOUDI collected particles in nine filters simultaneously. Each filter had a corresponding INP concentration, and each line in Fig. 2 (a) represents the sum of INP concentrations from 0.18 to 10 μm in each sample. On the whole, the measured total $N_{INP}$ spanned 4 orders of magnitude from $10^{-2}$ to $10^2$ L$^{-1}$ of standard air, and freezing was measured between -5 and -25 °C for dust events and between -15 and -28 °C for non-dust event. The total INP concentrations and freezing temperatures during dust and non-dust periods were significantly different, and the concentration increased by approximately 2 orders of magnitude during dust events at a given temperature, indicating that mineral dust particles are very efficient INPs in the immersion mode. For mineral dust particles, their freezing temperatures were similar. However, the variations in concentration between the samples were up to two orders of magnitude at a given temperature. The difference in INP concentrations was caused by dust particle properties including particle composition, particle loading, and the sample volume.

For comparison, two similar observations during dust events (Bi et al., 2019) and non-dust days (Chen et al., 2018b) in Beijing are illustrated in Fig. 2 (a). Bi et al. (2019) conducted a continuous measurement of INPs using a Continuous Flow Diffusion Chamber-Ice Activation Spectrometer (CFDC-IAS) from 4 May to 4 June 2018, and the $N_{INP}$ at -20 °C was 100 ± 80 L$^{-1}$ during dusty conditions, which agrees with the current study within one order of magnitude. For non-dust days, the concentrations in Chen et al. (2018b) are consistent with sample M4 in the present study.

The trend of $N_{INP}$ with temperature and particle size in mineral dust-dominated samples is depicted in Fig. 2 (b), where the size-resolved $N_{INP}$ ranged from $10^{-2}$ to $10^1$ L$^{-1}$ of standard air (see Table S2 in the SI). For each particle size, INP concentrations increased significantly with decreasing temperature. From another perspective, at a given temperature, $N_{INP}$ increased rapidly and then levelled when the particle size increased from 0.18 to 10 μm. This is explained by the fact that $N_{INP}$ depends not only on the activity of particles in a specific size range, but also on the total number concentration of the same size particles.

**3.2 Difference of INPs between two Transport Pathways**

In Fig. 3 (a), two distinct transport pathways, the northwest and north pathways, were identified during the sampling periods, based on air mass back trajectory analysis. These trajectories are consistent with the prevailing mineral dust transport pathways that affect the Beijing region (Huang et al., 2010;Sun et al., 2005). Combined with the geographical distributions of deserts, the northwest pathway passes through Gobi Desert and Kubuqi sandy desert in northwest China, while the north route passes through Hunshadake, Horqin and Hulun Buir sandy lands in north-eastern China. These two pathways shown in Fig. 3 are based on seven events. Four samples (M6, M7, M8, D7) originated from the northwest region and three samples (M3, M5, D6)





were from the northern area. Sample D7 is not included in Fig. 3 (c) because the size distribution data are partially missing, and surface area could not be fully derived.

It is clearly shown in Fig. 3 (b) that the total INP concentrations of samples that followed the same pathway were very similar in their freezing behavior. Although the concentrations in the two pathways have similar initial and final freezing temperatures, there was a significant difference of an order of magnitude in the intermediate temperatures (-16 ~ -11 ℃).

To compare the ice nucleation activity of different samples, ice active site density, $n_s(T)$, was calculated and is shown in Fig. 3 (c). Similar to $N_{INP}$ in Fig. 3 (b), the gross $n_s(T)$ values were similar in trajectories that followed the same route, but differed in trajectories between the two major pathways. This suggests that the mineral components differed between the different source regions; the ice nucleation activity of particles from the northwest was higher than that from the northern region. The surface area concentrations of the coarse mode particles ($D_p \geq 1.0$ μm ) were also higher in the northwest pathway (see Fig.

S2 in the SI).

Both $N_{INP}$ and $n_s(T)$ in the northwest pathway were higher than those in north route, suggesting that dust from northwest China deserts has a higher freezing efficiency. Previous studies have shown that Chinese deserts have distinct zoning characteristics; The north-western deserts are characterized by relatively higher amount of feldspars, while in the northern sandy lands, quartz mineral is more common (Zhao, 2015). The two dust sources in this study are consistent with these two

desert regions. The higher INP activity of the northwest pathway may be related to higher content of feldspar mineral, which is known as an important nucleator when temperatures higher than -20 ℃ (Atkinson et al., 2013).

### 3.3 Surface Ice Active Site Density of Dust Particles

Figure 4 (a) and (b) compares the gross and size-resolved (four size classes, $D_{50} = 5.6, 3.2, 1.8 \ and \ 1.0$ μm) $n_s(T)$ values from this study with other ambient samples dominated by mineral dust, respectively. In the present study, the gross $n_s(T)$

values span 4 orders of magnitude from $10^4$ to $10^8$ m$^{-2}$ at temperatures between -5 and -25 ℃. The $n_s(T)$ for $D_{50} = 5.6$ μm suggests higher nucleation activity, while the other three size classes show similar freezing performances. Price et al. (2018) reported airborne ice nucleating particles in the dusty tropical Atlantic, which is near the Sahara Desert in Africa (referred to as P18 from here on). Reicher et al. (2019) characterized the properties of size-segregated mineral dust sampled during dust events in the Eastern Mediterranean (referred to as R19 from here on), and found that the $n_s(T)$ values increased with particle

size.

The gross $n_s(T)$ values of most samples in this study are in agreement with P18, although a few samples are more active than P18 at higher temperatures (above -15 ℃). The measured temperature-dependency in this study is consistent with that observed in R19 in four size classes. The difference in the freezing temperature range between the two studies, higher in this study (-25 to -5 ℃) than in R19 (-35 to -20 ℃), is due to the droplets' volume (0.5 nL in R19, in contrast to 1 μL in the present study).

Larger droplets tend to freeze at higher temperatures because they contain a broader spectrum of nucleation active sites (O'Sullivan et al., 2014). Overall, we demonstrated that despite the fact that the mineral dust originated from different sources,



and experienced varying atmospheric transport and aging processes, both the gross and size-resolved ice nucleation activity showed great similarities over a wide range of temperatures (-25 ~ -5 ℃), which is consistent with the conclusions of previous studies (Niemand et al., 2012;DeMott et al., 2015;Kaufmann et al., 2016;Price et al., 2018;Reicher et al., 2019).

There are three possible explanations why some samples in this study were more active than the measurements in the two other studies presented in Fig. 4 (a) and (b). First is the content of feldspar in mineral dust. Nickovic et al. (2012) suggested that Asia has the highest feldspar content (up to 28%), much higher than that in central Sahara (~16%). Although not all feldspars are effective INPs (Harrison et al., 2016), feldspar content is thought to be an important factor. Some of the most efficient samples in this study were mainly from northwest China as shown in Sect. 3.2, where feldspar content may be higher than

China's north deserts and Africa's deserts (research regions in P18). Note that this is only a possible conjecture based on very limited evidence, and more field studies are needed. Secondly, in light of the difference in the vertical distributions of dust particles, the concentration of larger particles near the ground is higher (Maki et al., 2019). The near-surface-collected samples have a higher fraction of larger size particles, hence they may show more efficient INP activity than the aircraft measurements. Finally, another possible cause for the variability in $n_s(T)$ values above -15 ℃ may be the contribution of biological materials,

which are considered to be more active INPs (Murray et al., 2012;Hoose and Mohler, 2012;Kanji et al., 2017). Mineral dust dominated particles in our study were collected in the spring, when the plants germinate, grow and bloom. During the transport process, the air masses passed above grasslands, forests and green cities, so that mineral dust may mix with or serve as carrier of other more active species, possibly bacteria, fungi, pollen, plant materials and other microorganisms (Christner et al., 2008a;Pratt et al., 2009;Creamean et al., 2013;Boose et al., 2016a;Maki et al., 2018;Tang et al., 2018;Gat et al., 2017;Mazar

et al., 2016). It has been shown that fungal ice-nucleating proteins preferentially bind to and confer their ice-nucleating properties to mineral dust particles (O'Sullivan et al., 2016). In another study, O'Sullivan et al. (2014) found that a major fraction of INPs stemmed from biogenic components in the soil for temperatures above −15°C.

### 3.4 Contribution of Heat-Sensitive INPs

Previous studies have suggested that biological materials may attach to or mix with dust particles and play an important

contribution to INP populations during dust events (Pratt et al., 2009;Creamean et al., 2013;Boose et al., 2016a;Mazar et al., 2016;Gat et al., 2017). Because biological ice nucleation is mainly induced by proteinaceous components, inactivation by heat treatment is considered as a common way to identify biological nucleation activity (Christner et al., 2008a;Christner et al., 2008b;Garcia et al., 2012;O'Sullivan et al., 2014). In this study, heat-resistant INPs represent those particles that can initiate freezing after heat treatment (heated to 95 ℃ for 30 min). The suspensions of five size classes ( $D_{50} =$

$10, 5.6, 3.2, 1.8\ and\ 1.0$ μm) were measured before and after heat treatment to calculate original and heat-resistant $N_{INP}$, respectively. We subtracted heat-resistant $N_{INP}$ from the original concentration to evaluate the contribution of heat-sensitive INPs, which are mainly considered to be proteinaceous biological materials at warm temperatures.

Figure 5 illustrates the average proportion of heat-resistant and heat-sensitive INPs for five different size classes at three temperatures: -10, -15, and -20 ℃, derived from 12 sets of samples. At higher temperatures, heat-sensitive INPs dominated,





with 81 ± 12% at -10 °C and 70 ± 15% at -15 °C. At -20 °C, heat-resistant INPs increased to 62 ± 21%, in comparison with 19 ± 12% at -10 °C (see Table S3 in the SI). In general, the contribution of heat-sensitive INPs is similar for different particle size classes at a given temperature (see Table S4). This proportion is related to the balance of biological materials and active mineral dust. Microorganisms may be more common in larger particles due to larger surface area (Gong et al., 2020). On the other hand, larger particles may contain more efficient mineral species, such as feldspars (Margaret et al., 1994), which are expected

to be heat-resistant.

Figure 6 compares the freezing properties before and after heat treatment, based on those samples (with $D_{50} \geq 1.0\ \mu m$) originated from northwest and north pathways. The most prominent INP type above -10 °C were lost completely in north and northwest samples upon heating. In north samples, these INPs were less common than in northwest samples to begin with, a difference that extended to almost an order of magnitude in some of the cases. For example, $N_{INP}$ near temperature at -10 °C.

This is probably the reason why samples from the north pathway were less sensitive to heat treatment in general. It is then interesting to compare $n_s(T)$ curves of north and northwest samples in Fig. 6 (b), which demonstrated similar nucleation activity (within an order of magnitude, see Fig. S3 in the SI) after heating. This may suggest that after heat-sensitive INPs was removed, the two transport pathways are now dominated by similar material, which is probably mineral dust.

Overall, heat-sensitive components dominated the freezing at higher temperatures (above -15 °C), and their contribution

decreased pronouncedly at lower temperatures. Recently, Harrison et al. (2019) found that some quartz samples are sensitive to aging in aqueous suspension. We can't rule out the presence of quartz, or determine the fraction of specific quartz in heat-sensitive INPs at -20 °C. The higher proportion of heat-sensitive INPs at $D_{50} = 1.0\ \mu m$ at -20 °C may be attributed to the inactivation of quartz. However, quartz cannot be activated at -10 °C, and we can therefore conclude that a great proportion of INPs originated from heat-sensitive sites, mostly attributed to proteinaceous biological materials. The assumption of biological

contribution for higher nucleation activity above -15 °C was verified also in Sect. 3.3.

### 3.5 Parameterizations of Size-Resolved Ice Nucleation Active Site Densities

By combining the observational dataset in our study with those in R19, the new $n_s(T)$ parameterizations for size-resolved ice nucleation active site densities of dust particles are calculated by an exponential function:

$$n_s(T) = \exp(a * T + b)\ (m^{-2}) \tag{5}$$

Where $T$ is the temperature in Celsius, $a$ and $b$ are coefficients given in Table 2. These new parameterizations broaden the

temperature range spanning from -35 to -6 °C.

Four sizes, $D_{50} = 5.6, 3.2, 1.8\ and\ 1.0\ \mu m$, measured in both studies were considered for parameterization. The parameterizations for size-resolved dust particles are displayed in Fig. 7 (a) and their coefficients are given in Table 2. Here, the submicron fit line, marked in Fig. 7 (a) by the solid blue line, was derived from the dataset of $D_{50} = 0.6, 0.3\ \mu m$ in R19 and $D_{50} = 0.56, 0.32, 0.18\ \mu m$ in this study, representing the $n_s(T)$ of submicron desert dust. Note that we only present here

the cases dominated by mineral dust rather than all the data.





An explicit size dependent freezing efficiency was observed. The difference between the fit line of $D_{50} = 5.6$ μm and the submicron line were approximately 1 to 2 orders of magnitude, while the three fit lines of $D_{50} = 3.2, 1.8 \text{ } and \text{ } 1.0$ μm were almost overlapping, and located between the first two lines. In this case, the $D_{50} = 3.2, 1.8 \text{ } and \text{ } 1.0$ μm lines were integrated into one, 1.0 ~ 3.2 μm fit line, shown in Fig. 7 (b).

A comparison between the newly calculated parameterizations and those from previous studies (Niemand et al., 2012;Atkinson et al., 2013;Niedermeier et al., 2015;Reicher et al., 2019;Harrison et al., 2019) is presented in Fig. 7 (b). The parameterization in Niemand et al. (2012) is consistent with the 5.6 μm fit line (solid red line), and is close to the 1.0 ~ 3.2 μm fit line (solid green line) for temperatures lower than -17 ℃. The size of the surface-collected dust samples added into the chamber they used was less than 1 to 5 μm. Moreover, following a milling or sieving process, surface-collected samples may lead to higher

$n_s(T)$ than airborne samples (Boose et al., 2016b).

For different K-feldspar content predictions, an overlap of the new fit lines with previous studies is observed in the temperature range from -25 to -15 ℃, particularly for large size particles (5.6 μm and 1.0 ~ 3.2 μm), which contain more high-activity minerals (Boose et al., 2016b). Conversely, the submicron fit line coincides exactly with the 12% quartz parameterization by Harrison et al. (2019) at temperatures lower than -25 ℃. This phenomenon can be explained by the fact that quartz is more

active than clay minerals, thus may dominate the freezing ability in the smallest dust size fractions (Boose et al., 2016b;Reicher et al., 2019).

The parameterizations for supermicron and submicron particle size ranges in the lower temperature range (< -20 ℃) developed by R19 used the same data points as the present study. However, no size-resolved information for the higher temperature range was available. Combined with our dataset, a set of representative size-resolved parameterizations with a wide and

atmospherically relevant temperature range (-35 to -6 ℃) were derived, highlighting the importance of INPs size data.

The airborne particles, which were collected during dust-dominated events are composed of a complex mixture of various mineral components (e.g. feldspar, quartz, clay, and calcite), varying particle sizes, biological materials, and anthropogenic fine particulate matter. Its ice nucleation activity is determined by all components, and dominated by the most active substance. That is why these new parameterizations located between the K-feldspar and quartz parameterizations are more active at

temperatures higher than about -10 ℃. Overall, our new parameterizations can efficiently reflect actual atmospheric conditions.

## 4. Conclusions

The ice nucleation activities of size-resolved airborne East Asian dust particles in the immersion mode were investigated for the first time. Compared to non-dust event, the total INP concentrations during the East Asian dust events increased by approximately two orders of magnitude, and ranged between $10^{-2}$ and $10^{2}$ L$^{-1}$ of standard air at temperatures between -25 and

-5 ℃. The gross surface ice active site density, $n_s(T)$, spanned 4 orders of magnitude from $10^4$ to $10^8$ m$^{-2}$ at the temperature range of -25 ℃ to -5 ℃. Based on air mass back trajectory analysis, dust particles transported from China's northwest and



northern deserts have different ice nucleation efficiencies, indicating that dust particles from the northwest deserts may contain more active minerals.

An explicit size dependence of both INP concentration and surface ice active density was observed for Asian dust samples.
The nucleation efficiency of dust particles increased with increasing particle size, while the concentration first increased rapidly with particle size, and then levelled. This is due to the dependence of $N_{INP}$ on the common effect of the activity of individual size particles and the total number concentration of same size particles.

The gross and size-resolved $n_s(T)$ values were derived, and compared with recent studies. The results suggest that both the population and size-resolved ice nucleation activities of natural mineral dust particles from East Asia, North Africa, and Eastern
Mediterranean are relatively uniform, implying that the freezing properties of dust particles from global deserts are similar, as was found in previous studies (Boose et al., 2016b;Kaufmann et al., 2016;Price et al., 2018;Reicher et al., 2019).

During the East Asian dust events, the average contributions of heat-sensitive INPs at three temperatures, -10, -15, and -20 °C, were 81 ± 12%, 70 ± 15%, and 38 ± 21%, respectively. This result not only emphasizes the important role of biological materials during Asian dust transport, but also explains the phenomenon of higher INP activity at relatively warm temperatures
(above -15 °C) compared with the measurement closer to the desert source (P18). In addition, we found that the contribution of heat-sensitive INPs in different particle size classes was similar, which could be attributed to the abundance of adherent biological materials and active mineral components in the dust particles.

A new set of size-resolved parameterizations based on the field observations in this study and R19 were developed, which are valid in an extended temperature range spanning from -35 to -6 °C, characterizing the ice nucleation properties of size-resolved
mineral dust particles. The size of the particles controls their atmospheric lifetime, transport distance, and interactions with clouds, as larger particles sediment more quickly. Supermicron particles detected at high altitudes are much more abundant than expected by sedimentation theory alone (Ryder et al., 2018), emphasizing the importance of larger particles. Larger particles are more active INPs, as particle size reflects the mineral composition to a large extent. In both field observations and laboratory experiments it is easier to obtain particle size distributions than mineralogical compositions. Due to the single
requirement of particle size distributions as input for our model, the new particle size-based parametrizations can be widely applied in models, and help better characterize and predict INP concentrations related to natural mineral dust, especially related to long-range transport.

**Data availability.**

The data presented in this article can be accessed through the corresponding author Z. Wu (zhijunwu@pku.edu.cn).

**Supplementary Information**

The Supplementary Information related to this article is available online



**Author contributions.**

ZW designed and led the experiments. JC and JC collected the samples during dust events. JC, JC, XF and MH maintained the apparatuses and collected the dataset. The measurements and analysis of field samples were performed by JC under the
guidance of ZW and with the help of JC. All authors discussed the results and contributed to the writing of this paper. JC prepared the manuscript with contributions from all co-authors. ZW, JC, NR, YR and MH proofread and helped improve earlier versions of the manuscript.

**Competing interests.**

The authors declare that they have no conflict of interest.

**Acknowledgements.**

We gratefully acknowledge the NOAA Air Resources Laboratory (ARL) for the provision of the HYSPLIT transport and dispersion model and the MeteoInfoMap for the visualization and analysis used in this paper.

**Financial support.**

This work is supported by the following projects: National Natural Science Foundation of China (41875149, 91844301 and
385  4191101414).

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

Table 1. Overview of the 14 sets of samples collected during East Asian dust events in 2018 and 2019.

| Sample ID | Date | Weather condition | Total sampled volume (standard m$^3$) | PM$_{10}$ average (μg m$^{-3}$) | PM$_{10}$-peak value (μg m$^{-3}$) | INP concentration (L$^{-1}$) at -16 °C |
|---|---|---|---|---|---|---|
| M1 | 20180328 | Dust | 15.15 | 1145 ± 174 | 1521 | 17.36 |
| M2 | 20180328 | Dust | 24.16 | 252 ± 75 | 560 | 1.54 |
| M3 | 20180329 | Dust | 8.73 | 197 ± 11 | 221 | 0.69 |
| M4 | 20180329 | Non-dust | 36.48 | 120 ± 48 | 274 | 0.01 |
| M5 | 20180501 | Dust | 38.47 | 104 ± 59 | 260 | 0.42 |
| M6 | 20180505 | Dust | 12.37 | 397 ± 156 | 629 | 8.81 |
| M7 | 20180505 | Dust | 24.08 | 224 ± 52 | 354 | 4.87 |
| M8 | 20180526 | Dust | 8.94 | 270 ± 39 | 363 | 8.09 |
| D2 | 20190403 | Dust | 38.27 | 179 ± 50 | 263 | 1.49 |
| D3 | 20190405 | Dust | 15.06 | 465 ± 130 | 647 | 7.22 |
| D4 | 20190405 | Dust | 39.05 | 215 ± 34 | 266 | 5.56 |
| D5 | 20190417 | Dust | 38.19 | 163 ± 128 | 702 | 3.64 |
| D6 | 20190418 | Dust | 35.07 | 74 ± 34 | 214 | 0.89 |
| D7 | 20190512 | Dust | 39.01 | 116 ± 137 | 734 | 3.73 |






Table 2. Parameterization coefficients for different size classes based on the combined dataset in this study and R19 (Reicher et al., 2019).

| Fit line | Coefficients | $R^2$ | Valid T range (°C) |
|---|---|---|---|
| 5.6 μm-fit line | a= -0.441, b= 10.635 | 0.96 | -35 ~ -6 |
| 3.2 μm-fit line | a= -0.428, b= 9.592 | 0.95 | -35 ~ -6 |
| 1.8 μm-fit line | a= -0.425, b= 9.506 | 0.95 | -35 ~ -6 |
| 1.0 μm-fit line | a= -0.423, b= 9.390 | 0.89 | -35 ~ -6 |
| Submicron-fit line | a= -0.480, b= 5.926 | 0.76 | -38 ~ -10 |
| 1.0 ~ 3.2 μm-fit line | a= -0.425, b= 9.496 | 0.93 | -35 ~ -6 |

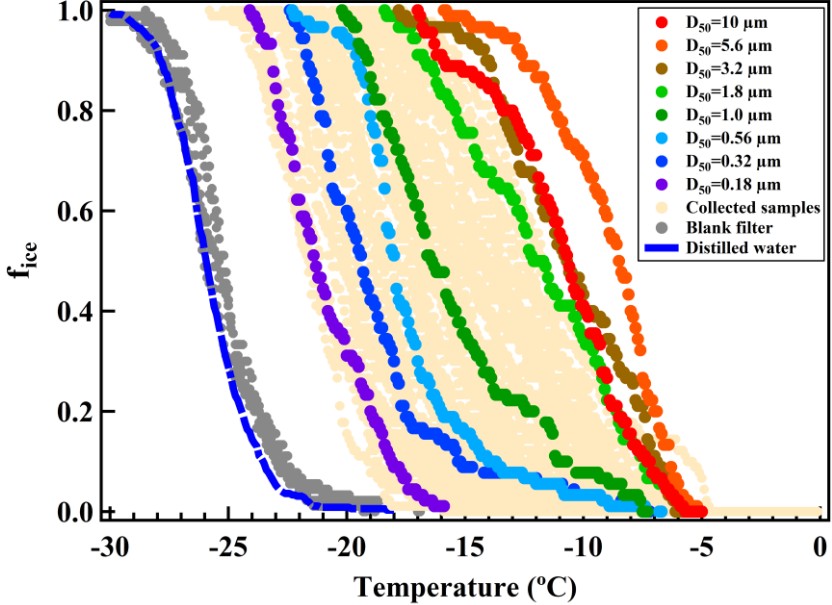

Figure 1. Frozen fraction curves of ambient particles during mineral dust events. The collected filter measurements are marked in pale yellow (named "Collected samples"), together with background $f_{ice}$ curves determined from blank filters (solid grey circles) and distilled water (dashed blue line). A set of filters ranging from 0.18 to 10.0 μm are marked with different colors to illustrate the frozen fraction curves for size-resolved particles. The set of size-resolved filters (sample M1) is part of the collected samples.





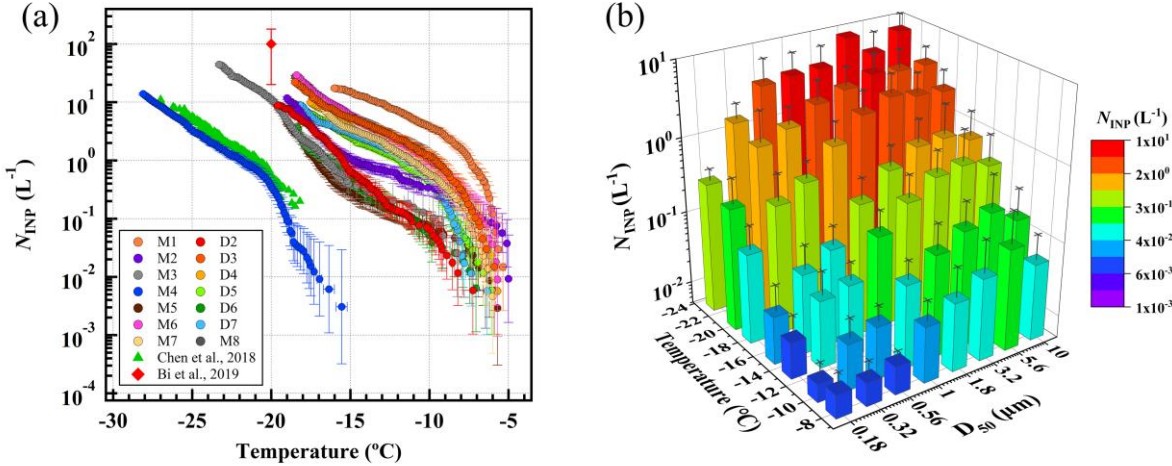

Figure 2. INP concentrations ($N_{INP}$) measured during the different events. (a) $N_{INP}$ of 14 sets of samples are plotted as a function of temperature (solid colored circles). The total INP concentrations in each event are represented by the different colors. $N_{INP}$ of non-dust days (solid green triangles) measured in Chen et al. (2018b) and dust event (solid red diamond) measured in Bi et al. (2019) are shown for comparison. Error bars of the INP concentrations indicate the 95% confidence intervals. (b) Size-resolved $N_{INP}$ versus temperature and particle class. Error bars represent the standard deviations in $N_{INP}$. Uncertainty in temperature values is 0.4 °C.



Figure 3. Comparison of INPs originating in northwest and north pathways. (a) Air mass trajectories for the different sampling events based on 72-h back trajectories. (b) INP concentrations ($N_{INP}$), and (c) ice nucleation active site density ($n_s(T)$) compared for the northwest and north pathways, marked by red and blue, respectively.





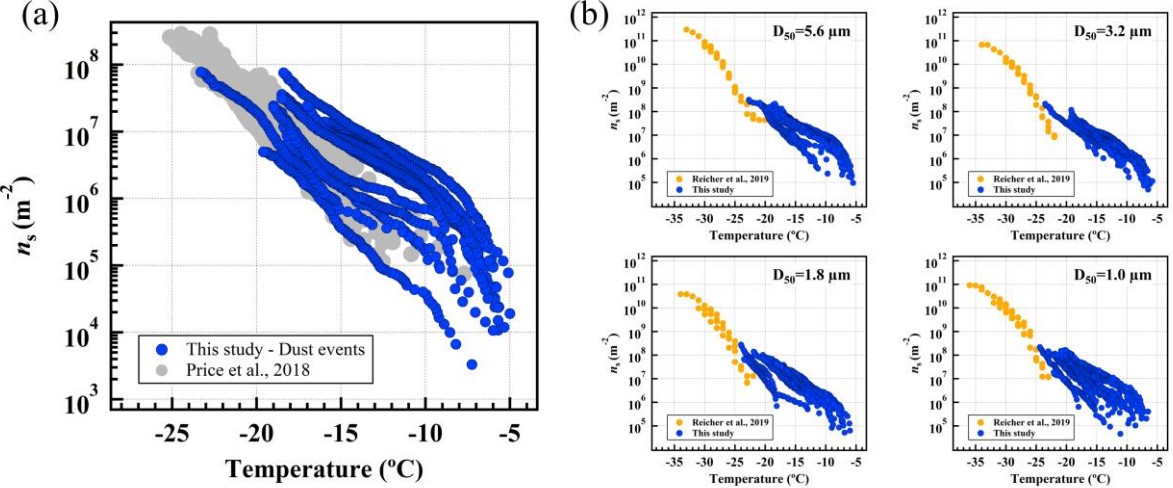

Figure 4. Comparison of Chinese mineral dust with African and East Mediterranean mineral dust. (a) Gross surface ice active site density ($n_s(T)$) from this study compared with aircraft measurements of mineral dust dominated samples in the eastern tropical Atlantic (solid grey circles) from Price et al. (2018). (b) Size-resolved $n_s(T)$ compared with mineral dust samples collected in the Eastern Mediterranean (solid yellow circles) from Reicher et al. (2019).

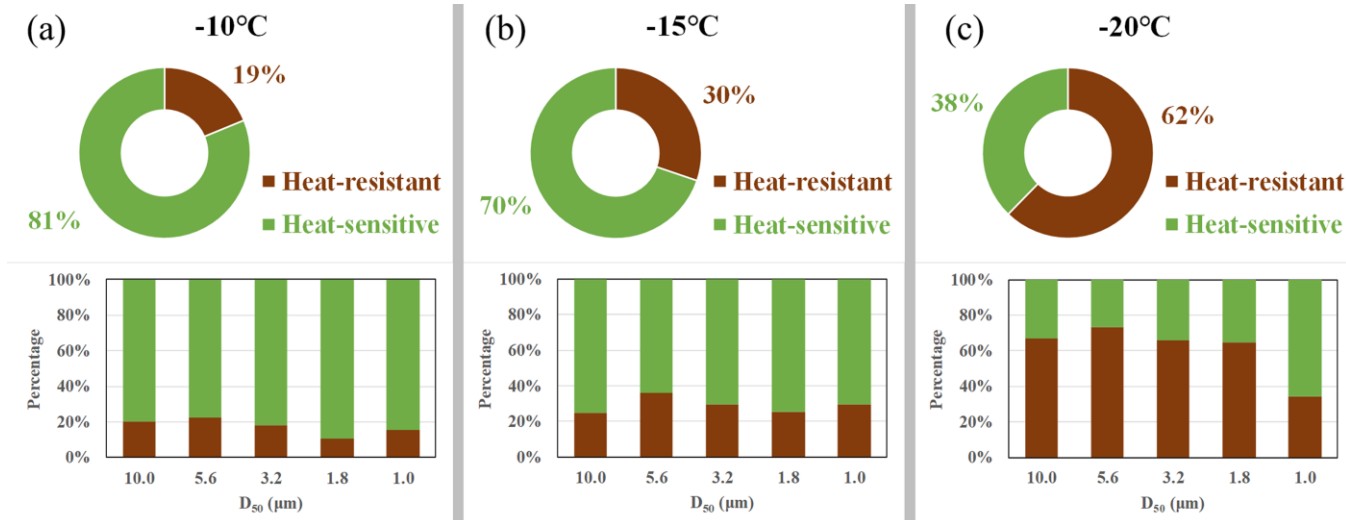

Figure 5. A summary of heating test results for various size classes. Characterisation of $N_{INP}$ after heat treatment is presented for three freezing temperatures, -10 °C (a), -15 °C (b), and -20 °C (c). The pie charts present the average contribution of heat-resistant and heat-sensitive INPs based on five analysed size classes ($D_{50} \geq 1.0$ μm), while the bar charts present the contribution for each size class ($D_{50} = 10, 5.6, 3.2, 1.8 \; and \; 1.0$ μm).



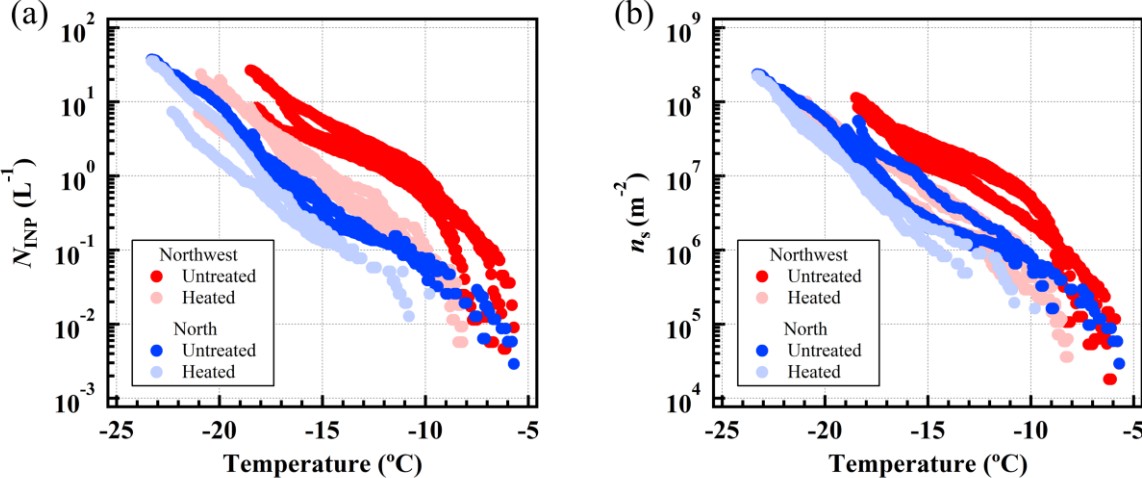

Figure 6. Effect of heat treatment on INP concentration and activity. (a) $N_{INP}$, and (b) $n_s(T)$ of northwest and north samples before (marked as "Untreated", solid red/blue circles) and after (marked as "Heated", solid light red/blue circles) heat treatment. The two figures present results with $D_{50} \geq 1.0$ μm.


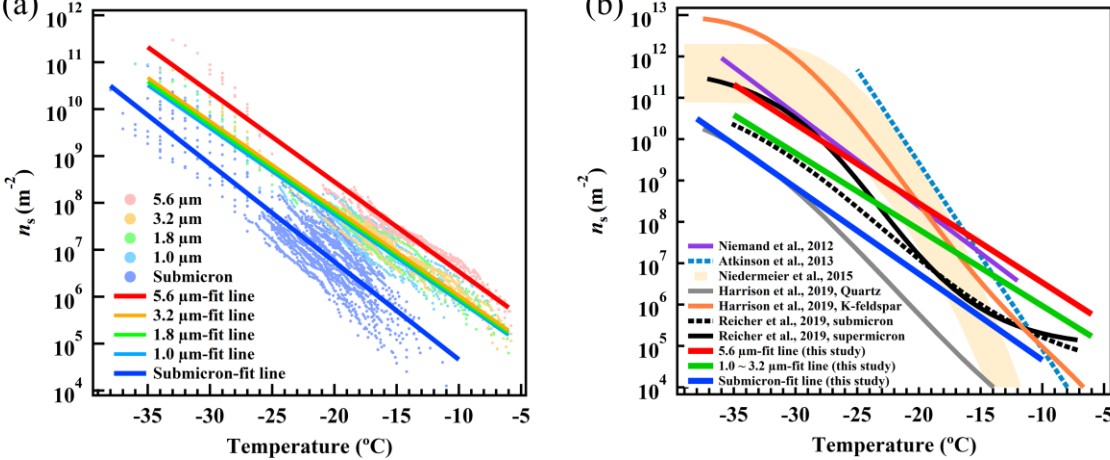

Figure 7. Parameterizations of ice nucleation active site densities $n_s(T)$ for different size classes. (a) Parameterizations developed from the combined dataset derived from this study and R19 (Reicher et al., 2019). Both the original data (colored solid points) and new fit lines (colored solid lines) are presented. (b) Comparison between the parameterizations derived from this study and those for desert dust (Niemand et al., 2012;Reicher et al., 2019) and single mineral dust components (Atkinson et al., 2013;Niedermeier et al., 2015;Harrison et al., 2019).
