# Peer review of "Size-Resolved Atmospheric Ice Nucleating Particles during East Asian Dust Events"

_Atmospheric Chemistry and Physics, 2020_

## Referee Comment (RC1) · Anonymous Referee #1 · 12 Oct 2020

**Access review to "Size-Resolved Atmospheric Ice Nucleating Particles during East Asian Dust Events" by Chen et al.**

Chen et al. present a laboratory investigation of the immersion freezing ice nucleation ability of filter-collected ambient Asian dust particles collected in Beijing.

Overall, I find that the topic of the manuscript fits well within the scope of ACP. This study extends previous studies on the immersion freezing ice nucleation ability of mineral dust particles to size-resolved measurements, and the experimental procedures and analysis are straight forward and sound. Based on the presented measurements a set of new parametrizations are developed that can predict the ice nucleation active surface site density of differently sized mineral dust particles at mixed-phase cloud conditions. While the results are mostly well presented and clear, the discussion of the parametrizations and comparison to previous parametrizations remains partly speculative. Therefore, I suggest the authors to address the below comments fore this manuscript is published in ACP

**General comments:**

- The discussion in Sect. 3.5 (in particular L321-335) to me reads somewhat confusing and in parts remains speculative. The present study does not present analysis of the mineralogical composition of the samples investigated. This makes it hard to follow the argumentation why the newly presented parametrizations should or should not follow previous parametrizations of desert dust samples that are based on samples of different but distinct mineralogical composition, but mostly on polydisperse aerosol particles (see Fig. 7b). Overall, it remains unclear whether the authors attribute the ice nucleation activity observed in the present study to particle composition or to particle size, when comparing to previous parametrizations. This section needs to be revised and more clearly structured upon revision.
- The authors suggest that the results help to understand the effect of chemical aging (e.g. L66, L73). However, specific aging mechanisms and or effects on the ice nucleation activity of the collected dust particles are not presented. I therefore suggest to remove the discussion of aging from the manuscript, unless a more comprehensive discussion of this topic is provided.

**Specific comments:**

- L17: Replace "warm" by "high"
- L24: Why is the upper limit -6 °C and not -5 °C, i.e. the upper limit of the presented immersion freezing experiments?
- L35: Delete "in-situ"
- L38: Change to: "... affects ice particle formation"
- L38-40: "simplified parametrizations" and "to accurately predict" seems contradictory; I suggest rephrasing this statement.
- L41: I suggest replacing "efficiency" by "ability", as the former implies some sort of time-dependence.
- L43: Add space before parenthesis here and on L44
- L46: Replace "and so on" by "such as"
- L47: "High content" and "increasing ratio" of what? Please specify.
- L49: Add Kumar et al. (2019)
- L53 : Add Kumar et al. (2019a), Zolles et al. (2015)
- L55 : Larger particles often… add Welti et al. (2009)
- L61: Do you mean "enhance the ice nucleation ability *to* higher temperatures"?
- L64-67: How does gravitational settling affect the dust transport and/or ice nucleation activity? Rephrase this statement.
- L69: Change to: "…in differently sized particles…"
- L71: Change to: "…activity of different…"
- L75: Change to: "…efficiency of Asian dust and its sensitivity to particle size, airborne…"
- L77: Change to: "…INP number concentration…"
- L78: Change "warm" to "high"
- L81: Climate models? Please specify.
- L86: Please specify the time resolution.
- L89: "A 8-stage…"

- L90: Change to: "We used stages 1 to 8 of the…at a flow rate of 30 L/min in this study." Move reference of Marple et al. (1991) to L89.
- L93: Delete "text"
- L104: Delete: "operated after careful temperature calibration"
- L108: How did the authors ensure that everything was washed of the filters? Was there any evidence from more sticky aerosol components, such as secondary organic material associated with the mineral dust particles?
- L120: Change to: "…concentration of ice active sites above…"
- L123: Change to: "is calculated as: …"
- L125: Change to: "…activity of samples with different aerosol particle size…"
- L127: Change to: "…(Vali et al. 2015) is calculated from the INP concentration as:…"
- L128: "and per droplet"?
- L128: Delete "based on the particulate matter information"
- L130: I do not follow this statement, please expand.
- L131: Delete "population"
- L133: Rephrase to. "Following the method of O'Sullivan et al. (2018)…"
- L137: Add "each particle size class"
- L143: Change to: "ultrafine condensation... »
- L184 : Change to : «…indicating different ice nucleating…"
- L193 vs. L195: Please write out "2" as "two" for consistency
- L227: Change "efficiency" to "ability"
- L230: "The higher…" Do the authors have any particle data to support this claim? Are there other studies that suggest the northwest pathway to be associated with a higher feldspar content?
- L233: Change to "Figures 4 (a) and (b) compare…"
- L236: Add a sentence a long the lines: "The ns values of this study are compared to literature values."
- L237: Change to "…desert in Africa"
- L243: Change to: "The difference in the temperature range between this study (…) add R19 (…) is due to the droplet volume…"
- L246: Change to: "…demonstrate that despite different origins of the dust samples investigated here and in R19, as well as the varying atmospheric transport..."
- L248: Delete "great" (it seems also a bit contradictory with the statement on L250).
- L253-255: Is this known? This statement should be supported by appropriate refernces
- L258: "The near-surface…" This is unclear, here you compare ns, which is normalized to particle size/surface area, or am I misunderstanding you here?
- L260: "…to be more active INPs" compared to dust?
- L262: Delete "green"
- L264-265: Can be reduced to 2-3 main references, as you detail these studies in Sect. 3.4.
- L270: Replace "population" by "number concentration"
- L272: Add: "…ice nucleation"
- L274: Delete space in front of D50
- L275: "and" should not be italicized
- L277: Replace "warm" by "high"
- L280: Do the indicated uncertainties correspond to standard deviations for the 12 samples? Please specify here and in the caption of Fig. 5.
- L290-293: This contradicts your hypothesis presented in Sect. 3.3. that different feldspar content contributes to different freezing abilities." There is not sufficient evidence provided to claim/suggest a difference in mineralogical composition between the two transport pathways. I suggest to completely leave this out and focus on the aspect of the biological fraction, where direct measurements and support is provided by your data. Please see my main comment above.
- L296: Change "can't" to "cannot"
- L304: Replace "Where" by "Here"
- L313: Replace "the first two lines" by "…between the lines of D50 = 5.6 µm and the ns curve for submicron particles."
- L317: "1.0 ~ 3.2 µm" I assume this line corresponds to the average of the D50 = 3.2, 1.8 and 1.0 µm lines which overlap in Fig. 7a, right? This should be specified in the text. I also suggest to replace "~" by "-" and chose a color that is distinctively different to any color used for the individual lines to avoid confusion.
- L319: "was less than 1 to 5 µm": Do you mean "below 5 µm"?

- L322: contain more highly ice active minerals"
- L323: Delete "exactly"
- L324: "-25 °C". From the figure it appears to be more likely "-29 °C".
- L324-325: "This phenomenon can…" If this was the case, why does your submicron parametrization deviate strongly from the quartz parametrization of Harrison et al. (2019) at higher temperatures? Is this because feldspar ice nucleation activity dominates at higher temperatures? This should be specified.
- L333: Replace "components" by "factors"
- L334-335: "…are mire active…" Compared to what? The Atkinson et al. (2013) K-feldspar line ?
- L335 : « Overall… » This statement seems misplaced and should be moved to Sect. 4
- L342-343: Please see my comment above. Your data suggest that the difference is mainly driven by a difference in the biological material present on the dust particles from the two transport pathways.
- L354: Replace "warm" by "high"
- L355: Are you trying to say that Asian dust has a higher abundance of biological material compared to desert dust?
- L362: "…emphasizing the importance…" I suggest to tune this down a little bit: "…potentially suggesting the importance of larger particles for cloud formation."
- L363: "as particle size reflects … » This statement should be support by references.
- L364-365: "Due to the single requirement…" Unclear what you mean, please rephrase.
- Fig. 3: Why are there two blue lines coming from the northwest pathway, i.e. lie on top of the red trajectories?
- Fig. 5: Include space between value and unit, i.e. "10 °C"

Kumar, A., Marcolli, C. and Peter, T.: Ice nucleation activity of silicates and aluminosilicates in pure water and aqueous solutions – Part 2: Quartz and amorphous silica, Atmospheric Chem. Phys., 19(9), 6035–6058, doi:https://doi.org/10.5194/acp-19-6035-2019, 2019a.

Kumar, A., Marcolli, C. and Peter, T.: Ice nucleation activity of silicates and aluminosilicates in pure water and aqueous solutions – Part 3: Aluminosilicates, Atmospheric Chem. Phys., 19(9), 6059–6084, doi:https://doi.org/10.5194/acp-19-6059-2019, 2019b.

Welti, A., Lueoend, F., Stetzer, O. and Lohmann, U.: Influence of particle size on the ice nucleating ability of mineral dusts, Atmospheric Chem. Phys., 9(18), 6705–6715, 2009.

Zolles, T., Burkart, J., Häusler, T., Pummer, B., Hitzenberger, R. and Grothe, H.: Identification of Ice Nucleation Active Sites on Feldspar Dust Particles, J. Phys. Chem. A, 119(11), 2692–2700, doi:10.1021/jp509839x, 2015.

---

## Referee Comment (RC2) · Anonymous Referee #2 · 10 Nov 2020

In this manuscript, a study is introduced for which size segregated airborne aerosol samples, sampled during desert dust events in China, were examined concerning the particles ability to act as ice nucleating particle (INP). Besides for the effect of size, also possible contributions from biogenic materials were examined.

The study was well done and is well presented. The results are very timely. I have a few concerns, but nothing really major. Major revisions are only recommended (instead of minor revisions) due to the number of comments. However, after addressing my below comments, the manuscript will very likely be suited for publication in ACP.

major remarks:

line 23-24: What is the advantage of a size dependent parameterization over a single

n_s curve that could be used with the total surface area of an aerosol? This could be discussed later in the text. Also, you could estimate which error would be done if one used such a single n_s curve, compared to your higher size resolved information, for which, however, more modeling capacity will be needed.

section 2.1: You do discuss losses in a MOUDI a bit. But I still wonder if the filters put on the MOUDI-stages did not change the collected sizes, as this is sensitive to the distance between the plates. Also, the filters are much larger than the collection area of a MOUDI - were the filters cut or how was this issue dealt with? And how were they kept in place?

line 149: It is clear that either diameters as measured by the SMPS needed to be adapted, or the aerodynamical diameters as measured by APS and as selected by MOUDI. But without checking literature thoroughly, I would have expected that geometrical diameter is the one most previous studies refered to, not aerodynamic diameter, when retrieving n_s. Please check this and give an estimation of the deviation this may cause.

line 170 ff: The description of the definition of the dust events needs to be adjusted: from Table 1, it seems that it was based on CMA observations, only. Otherwise M5, D6 and D7 would also not be dust events.

lines 175ff: Were results from M4 included in this study at all? If yes, how? If not, why mention it? And why are M1, M2, M3 and M4 described here explicitly, but not all the other samples? A sentence or two on the samples, even if they are only summarized in groups, would be helpful, so that for each sample its specifics are clearer (like the one found in line 214 - you can repeat some of that information there again, but also add it here).

line 178: The data shown in Fig. 1 in "pale yellow" - are they all for size segregated filters? If so, why is the set of colorfully depicted data well above these curves for the three larges particle sizes? If this colorfully depicted data-set is one with particularly

high ice activity, mention this explicitly.

Line 183-185: Based on f_ice curves, it cannot be judged which size class is more ice active. It is theoretically possible that there are just so many more (or less) particles in one size class than in others, so that this number is the overwhelming influence on the overall measured ice activity at that impactor stage, causing high (or low) f_ice. I suggest that you at least mention this in an additional sentence.

line 194-195: "For mineral dust particles, their freezing temperatures were similar." Do you mean that the dust curves were all similar? That is not true, and you should normalize to dust surface area first before you make comments on that, anyway. You even mention that in the next sentences. This sentence here needs to be deleted or revised.

line 203: Fig. 2 (b): Is this based on one sample, or an average from all? Mention this! And again, it might make more sense to show this for size-normalized data, but number is fine, here, too.

line 214: Why did you choose to only show results for this subset of samples? This needs to be justified in the text, otherwise the impression could arise that you did "cherry picking", i.e., only displaying those samples that fit what you want to see.

line 215: Is Sample D7 included in Fig. 3 (b)? To me, only 6 curves are visible in Fig. 3 (b), too. Please check, and if D7 is not included, mention this in the text.

line 217: The airborne INP concentration depends on getting dust suspended in the air. So even the same source region can yield different airborne concentrations for different conditions such as changing wind speed. Mention this additional restriction. Different shapes, however, really depict different types of INP. (Like, based on what you describe later, a biogenic component in the "red" samples.)

line 233: Concerning Fig. 4 (a) and (b), it seems as if here only 10 different data-sets are shown. Is that correct, and if yes, why were the other data not shown? This is in

line with my comment concerning line 214.

line 235: It would be more informative to mention the temperature span at a single temperature, as this overall span depends on the measurement method you use! (It varies with the amount of air you collect, the surface area (hence the size distribution) and, to a lesser extent, to the number of droplets you examine, but NOT on any characterization of the INP.)

line 286: Again, why is only a subset of all 14 sets of filter samples shown?

line 292-293: "This may suggest that after heat-sensitive INPs was removed, the two transport pathways are now dominated by similar material, which is probably mineral dust." I totally agree - but that makes any discussion of different feldspar contents, which you did above, futile. Please check the content of the text for consistency! Or, when you mention feldspar, already point out that this may not be important as the importance of the biogenic content will be discussed below.

line 299-300: What does that sentence refer to (in Sect. 3.3)? Please explain what you mean.

line 223ff: My advice is to not overinterpret such observations. There are measurement uncertainties on all of these curves, and there are different approaches. The $n\_s$ derived in a lab-study from mineral dust samples refers to the surface area of dust particles, only, while in your study, you naturally have to refer to the surface area of the total aerosol. Also, you used the aerodynamic diameter as the reference, while this, to my understanding, was not the case in the other studies. So please be careful when discussing such details.

line 333-335: How does that fit with the fact that you see such a high biogenic / proteinaceous fraction being responsible for the ice activity at higher temperatures??? Also: The deviation can be seen at high temperatures, where your fits are much above the mineral-dust parameterizations - mention that explicitly. Also: Fitting a straight line

over such a broad T-range might be misleading.

line 340-341: As said above, this temperature range rather characterizes your method than saying anything about the INP. Give the span at a single temperature, as this signifies who different your different samples were.

line 346: "the common effect of the activity" - what do you mean by that. This could be elaborated somewhat more, maybe even in an additional sentence.

line 350: Is that really what you find. You argued with different feldspar content at some point, then with different biogenic content, and now you summarize all this in saying "all desert dusts are the same". Make the message of your text consistent throughout the manuscript.

line 362ff: "Larger particles are more active INPs, as particle size reflects the mineral composition to a large extent" - This is not necessarily true. If larger particles have a higher n_s, then it's true, but in general larger particles are more ice active because they have a larger surface area. Formulate this with more care.

Technical issues and minor remarks:

line 29: "relatively high temperatures" - say more precisely what you mean by that.

line 37: What exactly do you mean by "mid-level clouds". The use of "mixed-phase clouds" (as in the next sentence) seems more appropriate here.

line 52: "supplement for feldspar" - it is not clear what you mean, here. That the two always occur together? That would not be correct, as feldspars are weathered clay minerals, and quartz is not a clay mineral. Please check and reword.

line 91: "stages . . . were detected" - wrong wording, needs to be changed.

line 108: Are you aware of the fact that the use of ultrasonic waves may change the structure of proteins and therewith change their functionality? (see. e.g., DeLeo et al., 2016) At least mention this in your text, so that future readers know about this issue

when they consider repeating what you did.

line 115: change "will not be expected and" to "is not".

line 120: This is the first time that active sites are mentioned, so you may want to add a few words on explaining what you mean by that.

line 138: "Gross" seems a bit misplaced here. I suggest to use a different word. Or, as you use "gross" more often, what you mean by that.

line 287: Change "originated" -> "originating".

line 289: "For example, ðĺŚĄ_ðĺŘijðĺŚĄðĺŚČ near temperature at -10 °C." This is not a complete sentence - check and correct.

Fig. 2: "b" is missing in the legend for Chen et al. (2018b). Also, change the color either for the Bi et al. (2019) datapoint, and/or make it an open symbol (maybe an open star?), as it is difficult to discriminate between these data and those from D2.

Literature: De Leo et al., Effect of ultrasound on the function and structure of a membrane protein: The case study of photosynthetic Reaction Center from Rhodobacter sphaeroides, Ultrason. Sonochem. (2016), http://dx.doi.org/10.1016/j.ultsonch.2016.09.007 .

---

## Author Comment (AC1) · 12 Jan 2021

*The authors are grateful to the editor and referees for their careful reading and constructive suggestions that substantially help to raise the quality of our manuscript. Below we address each of the comments listed in blue font. Our answer is listed in black font and revised text is listed in green font. The number of lines in our answers is based on the revised manuscript, and the amendments were marked with a highlight in the revised version.*

**Referee #1:**

Chen et al. present a laboratory investigation of the immersion freezing ice nucleation ability of filter-collected ambient Asian dust particles collected in Beijing.

Overall, I find that the topic of the manuscript fits well within the scope of ACP. This study extends previous studies on the immersion freezing ice nucleation ability of mineral dust particles to size-resolved measurements, and the experimental procedures and analysis are straight forward and sound. Based on the presented measurements a set of new parametrizations are developed that can predict the ice nucleation active surface site density of differently sized mineral dust particles at mixed-phase cloud conditions. While the results are mostly well presented and clear, the discussion of the parametrizations and comparison to previous parametrizations remains partly speculative. Therefore, I suggest the authors to address the below comments fore this manuscript is published in ACP.

We appreciate the referee's affirmation and comments on our work. The comments were responded point-by-point in the following contents, and the manuscript was revised. We have made direct responses and substantial revisions which we believe properly address the referee's concerns.

**General comments:**

1. The discussion in Sect. 3.5 (in particular L321-335) to me reads somewhat confusing and in parts remains speculative. The present study does not present analysis of the mineralogical composition of the samples investigated. This makes it hard to follow the argumentation why the newly presented parametrizations

should or should not follow previous parametrizations of desert dust samples that are based on samples of different but distinct mineralogical composition, but mostly on polydisperse aerosol particles (see Fig. 7b). Overall, it remains unclear whether the authors attribute the ice nucleation activity observed in the present study to particle composition or to particle size, when comparing to previous parametrizations. This section needs to be revised and more clearly structured upon revision.

We thank the referee for this comment.

We did not investigate the quantitative mineralogical composition in this study, so that there is no solid evidence to explain the discrepancy in terms of mineral composition. Combining the comments of the two referees, we chose to be cautious in explaining the differences with other studies and to focus on the results determined by our experiment. This section was thoroughly rephrased in the revised manuscript:

"Figure 7 (b) compares our size-resolved parameterizations with those from previous studies, based on desert dust (Niemand et al., 2012;Reicher et al., 2019) and single mineral dust components (Atkinson et al., 2013;Niedermeier et al., 2015;Harrison et al., 2019). Niemand et al. (2012) measured surface-collected dust samples (less than 5 µm) to derive the parameterization (solid purple line), which is consistent with our 5.6 µm-fit line (solid red line), and is close to the 1.0 - 3.2 µm-fit line (solid dark green line) for temperatures higher than -17 ℃. For different K-feldspar content predictions, an overlap of the new fit lines with previous studies is observed in the temperature range from -25 to -15 ℃, particularly for large size particles (5.6 µm and 1.0 - 3.2 µm). At temperatures below -29 ℃, the submicron-fit line coincides with the 12% quartz parameterization by Harrison et al. (2019), but is 1 to 2 orders of magnitude higher above this temperature. The supermicron and submicron parameterizations developed by R19 agree within an order of magnitude with our three parameterizations in the lower temperature range (< -23 ℃). These two parameterizations underestimate the nucleation activity of large size particles (5.6 µm and 1.0 - 3.2 µm), and fit for the submicron particles for

temperatures higher than -20 ℃.

We note that the quantitative mineralogical composition was not investigated in this study, so that we cannot explain the discrepancy accurately in terms of mineral composition. On the other hand, while relatively minor, measurement and calculation uncertainties should be borne in mind when comparing our parameterizations with other curves as well. First, different experimental methods introduce measurement errors. A cold stage-based technique was applied in this study, while cloud simulation chamber (Niemand et al., 2012), laminar flow tube (Niedermeier et al., 2015) and many other cold-stage instruments (with varying size/volume droplets; Atkinson et al., 2013;Harrison et al., 2019;Reicher et al., 2019) were used to measure the activated fractions of tested particles/droplets at a given temperature. Then, the investigated particles came from various sources and underwent different processing, including airborne-collected, surface-collected (sieved or milled) samples, and single mineral dust components. Next, the calculation of $n_s(T)$ depends on a key parameter, particle surface area, which refers to the surface area of dust particles in laboratory studies, while refers to the surface area of total aerosol particles in this study and in R19. Furthermore, we adopted aerodynamic diameter to obtain $n_s(T)$, which underestimated the result (0.42 to 0.93 times) compared with that determined by the converted geometric diameter.

These airborne dust particles are composed of a complex mixture of various mineral components (e.g. feldspar, quartz, clay, and calcite), varying particle sizes, biological materials, and anthropogenic fine particulate matter. Its ice nucleation activity was determined by all factors, and dominated by the most active substance. Despite the uncertainties, it is certain that there is an explicit size dependent freezing efficiency over a large temperature range, and the contribution of biological materials to nucleation activity at T > -15 ℃ is highlighted. Compared with mineral composition-based parameterizations, the advantage of particle size-based curves is that we do not need to know the complex mineralogical composition of dust. Only the particle size distribution, a widely monitored

parameter, is required in size-based prediction. Furthermore, there is a vertical distribution of mineral dust in the atmosphere (Maki et al., 2019), implying potentially different contributions of these various size particles in cloud formation. Our size dependent parameterizations can provide more refined simulation and prediction in theory, which needs to be confirmed in further model studies." (L336-368)

2. The authors suggest that the results help to understand the effect of chemical aging (e.g. L66, L73). However, specific aging mechanisms and or effects on the ice nucleation activity of the collected dust particles are not presented. I therefore suggest to remove the discussion of aging from the manuscript, unless a more comprehensive discussion of this topic is provided.

We thank the referee for this suggestion. Specific measurements and analysis of chemical aging of the collected Asian dust particles are indeed not presented in the manuscript. We deleted the discussion of chemical aging to make the topic clearer. The sentences were rephrased:

"In fact, reference single mineral dust and surface-collected particles do not fully represent the actual dust transport process in the troposphere due to gravitational dust sedimentation, adsorption of biological materials, and other factors." (L64-66)

"…the role of Asian dust, especially after long-range transport, …" (L74)

**Specific comments:**

1. L17: Replace "warm" by "high".

Replaced (L17, L79, L292, L387).

2. L24: Why is the upper limit -6 °C and not -5 °C, i.e. the upper limit of the presented immersion freezing experiments?

The upper limit temperature was based on our experimental results. As you can see in Fig. 2(a), Fig. 3(b), Fig. 4(a), Fig. 6(a) and Fig. 7(a), few samples froze at -5 °C, whereas most samples nucleated from -6 °C. So that we define the upper limit of

the valid temperature range in this study is -6 °C, as given in Table 3.

3. L35: Delete "in-situ".

Deleted as suggested.

4. L38: Change to: "... affects ice particle formation"

Replaced (L38).

5. L38-40: "simplified parametrizations" and "to accurately predict" seems contradictory; I suggest rephrasing this statement.

The common goal of cloud studies is to accurately simulate and predict the occurrence and evolution of clouds in models. At the same time, concise and elegant parameterizations are also the pursuit of scientific researchers.

This sentence was rephrased for clarity:

"However, parameterizations characterizing INP activity are required to predict the occurrence and evolution of clouds, suggesting that there is a need for measurements of ice formation on different INP types." (L38-41)

6. L41: I suggest replacing "efficiency" by "ability", as the former implies some sort of time-dependence.

Replaced (L41).

7. L43: Add space before parenthesis here and on L44.

Added. Thanks for the referee's careful reading, and the full text has been checked.

8. L46: Replace "and so on" by "such as"

This sentence was rephrased (L45-46):

"Dust particles are mainly composed of clay minerals (including illite, kaolinite, chlorite, etc.), quartz, feldspar, calcite, and other mineral components."

9. L47: "High content" and "increasing ratio" of what? Please specify.

"High content" and "increasing ratio" of the clay minerals.

The statement was modified (L47-49):

"Clay minerals were widely investigated in ice nucleation studies (Mason, 1960;Eastwood et al., 2008;Pinti et al., 2012;Wex et al., 2014;Kumar et al., 2019a) due to their high abundance in mineral dust composition (Murray et al., 2012), especially after long-range transport (Leinen et al., 1994;Uno et al., 2009)."

10. L49: Add Kumar et al. (2019)

Added as suggested (L48).

11. L53: Add Kumar et al. (2019a), Zolles et al. (2015)

Added (L53).

12. L55: Larger particles often… add Welti et al. (2009)

Added (L56).

13. L61: Do you mean "enhance the ice nucleation ability to higher temperatures"?

We followed the comment and rephrased the statement:

"… extend the ice nucleation ability to higher temperatures" (L61-62)

14. L64-67: How does gravitational settling affect the dust transport and/or ice nucleation activity? Rephrase this statement.

We followed the comment and added more details:

"During dust transport, larger particles settle faster due to gravity, while smaller particles can remain lifted for longer period, thus possibly playing a different role in cloud formation (Kramer et al., 2020;Maki et al., 2019)." (L66-68)

15. L69: Change to: "…in differently sized particles…"

Changed (L70).

16. L71: Change to: "…activity of different…"

Changed (L72).

17. L75: Change to: "…efficiency of Asian dust and its sensitivity to particle size, airborne…"

Changed (L76).

18. L77: Change to: "…INP number concentration…"

Changed (L78).

19. L78: Change "warm" to "high"

Changed (L79).

20. L81: Climate models? Please specify.

The $n_s(T)$ parameterizations can be applied in regional and/or global climate models, such as the Single-column version of the Community Atmospheric Model version 5 (SCAM5, Neale et al., 2010), the Consortium for Small-scale Modeling (COSMO, Baldauf et al., 2011), and the global chemical transport model GEOS-Chem (Schill et al., 2020)

References:

- Baldauf, M., Seifert, A., Förstner, J., Majewski, D., Raschendorfer, M., and Reinhardt, T.: Operational convective-scale numerical weather prediction with the COSMO model: Description and sensitivities, Mon. Weather Rev., 139, 12, 3887–3905, doi:10.1175/MWR-D-10-05013.1, 2011.

- Neale, R. B., Chen, C-.C., Gettelman , A., Lauritzen, P. H., Park, S., Williamson, D. L., Conley, A. J., Garcia, R., Kinnison, D., Lamarque, J-.F., Marsh, D., Mills, M., Smith, A. K., Tilmes, S., Vitt, F., Morrison, H., Cameron-Smith, P., Collins, W. D., Iacono, M. J., Easter, R. C., Ghan, S. J., Liu, X., Rasch, P. J., and Taylor, M. A.: Description of the NCAR Community Atmosphere Model (CAM5.0), Tech. Rep. NCAR/TN-486-STR, NCAR, available at: http://www.cesm.ucar.edu/models/cesm1.0/cam/, 2010

- Schill, G. P., DeMott, P. J., Emerson, E. W., Rauker, A. M. C., Kodros, J. K., Suski, K. J., Hill, T. C. J., Levin, E. J. T., Pierce, J. R., Farmer, D. K., and Kreidenweis, S. M.: The contribution of black carbon to global ice nucleating particle concentrations relevant to mixed-phase clouds, Proceedings of the National Academy of Sciences, 117, 22705, 10.1073/pnas.2001674117, 2020.

21. L86: Please specify the time resolution.

    Changed to "…minute-level temporal resolution meteorological parameters…" (L87).

22. L89: "A 8-stage…"

    Changed to "An eight-stage…" (L90).

23. L90: Change to: "We used stages 1 to 8 of the…at a flow rate of 30 L/min in this study." Move reference of Marple et al. (1991) to L89.

    Corrected.

    "We used stages 1 to 8 of the MOUDI with cut-points ($D_{50}$) ranging from 10 to 0.18 μm in aerodynamic diameters at a flow rate of 30 L min$^{-1}$ in this study." (L92-93)

24. L93: Delete "text"

    Deleted.

25. L104: Delete: "operated after careful temperature calibration"

    Deleted.

26. L108: How did the authors ensure that everything was washed of the filters? Was there any evidence from more sticky aerosol components, such as secondary organic material associated with the mineral dust particles?

    We thank the referee for this comment, and added related information.

    We did an experiment of particle washing removal efficiency to ensure that

particles were washed off the filters, as depicted in Fig. R1.1. The tested filter was collected from a dust event in 2018, and all the extraction processes were the same as those described in the manuscript except for the extraction time. This Nuclepore filter was completely submerged in 20 mL double-distilled water (resistivity of 18.2 MΩ·cm at 25 °C) and was extracted by an ultrasonic shaker for 15 minutes to get the sample called "15 min - 1st" in Fig. R1.1. Then the filter was removed from the washed suspension and was immersed in a fresh 20 mL double-distilled water for a second extraction cycle to obtain the sample called "15 min - 2nd". The sample "15 min - 3rd" was produced similarly in a third extraction cycle.

[Figure]

Figure R1.1. Particle washing removal efficiency experiments. The brown, orange and light orange bars represent the frozen fraction ($f_{ice}$) of the solution at different temperatures after the first, second, and third ultrasonic treatment, respectively. As comparisons, the grey and blue bars are the droplet freezing experiments for blank filter and distilled water, respectively.

The freezing of the three samples indicated that most of the particles were extracted efficiently in the first cycle, which had higher frozen fractions at higher temperatures than rest of the samples. Indeed, some of the particles remained over the filter, but a longer extraction periods would not impact the freezing results, since there was only minor overlap between their freezing temperatures. Therefore,

15-minute ultrasound treatments for twice (i.e., 30 minutes) can wash all ice active materials off the filters. As for the more sticky aerosol components, we think they were also eluted into the aqueous solution according to our results.

The above content was detailed in the revised manuscript and supplementary information.

"The extraction process lasted 30 minutes, … (see Supplementary Information, referred to as SI from here on, for more details)." (L110)

27. L120: Change to: "…concentration of ice active sites above…"

Changed (L123).

28. L123: Change to: "is calculated as: …"

Changed (L127).

29. L125: Change to: "…activity of samples with different aerosol particle size…"

Changed (L129).

30. L127: Change to: "…(Vali et al. 2015) is calculated from the INP concentration as:…"

Changed (L131).

31. L128: "and per droplet"?

We gratefully thank the referee for the careful reading.

In Eq. (3), parameter $A$ should be the total surface area of the particles per unit volume of sampled air. The error in the text has been corrected. We make sure the calculations and results are correct.

"where $A$ is the total surface area of the particles per unit volume of sampled air, …" (L132)

32. L128: Delete "based on the particulate matter information"

Deleted.

33. L130: I do not follow this statement, please expand.

We followed the comment and added some detailed explanations.

A principal source of uncertainty in the experiments stems from the representativeness of testing droplets for the total suspension, especially for the scenario that only a minor fraction of droplets contains ice active particles. We added more discussion about this statement as follows:

"The INPs are scarce in the air, thus their number presented in the washing suspension is usually small. These small volume droplets (1 µL) may not contain a representative number of particles, and the number of examined droplets is limited (90 droplets)." (L136-138)

34. L131: Delete "population"

Deleted.

35. L133: Rephrase to. "Following the method of O'Sullivan et al. (2018)…"

This statement has been reworded to "Following the method of O'Sullivan et al. (2018) and Barker (2002), …" (L139)

36. L137: Add "each particle size class"

Added (L143).

37. L143: Change to: "ultrafine condensation... »

Changed (L150).

38. L184: Change to: «…indicating different ice nucleating…"

Changed (L193).

39. L193 vs. L195: Please write out "2" as "two" for consistency

It was uniformly written as "2" (L202). The same question was reworded as "2 orders of magnitude" in Line 372 for consistency.

40. L227: Change "efficiency" to "ability"

Changed (L241).

41. L230: "The higher…" Do the authors have any particle data to support this claim? Are there other studies that suggest the northwest pathway to be associated with a higher feldspar content?

We did not measure the mineral compositions in this study and no reliable evidence to support this view. Combined with other comments from the two referees, we decided to downplay the mineral fractions and focus on biological materials, which has been confirmed by reliable measurement results.

Here, the sentences have been modified:

"On the one hand, previous studies have shown that Chinese deserts have distinct zoning characteristics; The north-western deserts are characterized by relatively higher amount of feldspars, while in the northern sandy lands, quartz mineral is more common (Zhao, 2015). The two dust sources in this study are consistent with these two desert regions. On the other hand, the high ice nucleation activity above -15 °C may be attributed to the attached biological materials on the dust particles (Tang et al., 2018). The reasons for the different INP activity of two pathway samples are discussed in detail below." (L241-246).

42. L233: Change to "Figures 4 (a) and (b) compare…"

Changed (L248).

43. L236: Add a sentence along the lines: "The ns values of this study are compared to literature values."

Added (Line 252).

44. L237: Change to "…desert in Africa"

Changed (L253).

45. L243: Change to: "The difference in the temperature range between this study (…) add R19 (…) is due to the droplet volume…"

Changed (L258-259):

"The difference in the temperature range between this study (-25 to -5 °C) and R19 (-35 to -20 °C) is due to the droplet volume (0.5 nL in R19, in contrast to 1 μL in the present study)."

46. L246: Change to: "…demonstrate that despite different origins of the dust samples investigated here and in R19, as well as the varying atmospheric transport..."

Changed (L261-262):

"…demonstrate that despite different origins of the dust samples investigated here and in R19, as well as the varying atmospheric transport processes, …"

47. L248: Delete "great" (it seems also a bit contradictory with the statement on L250).

Deleted (L262)

We also qualified the "similarity" into "within 1 or 2 orders of magnitude" (L262) and rephrased the statement in L265:

"However, some samples in this study were more active than the measurements in the above two studies. Three hypothesizes are proposed to explain the possible reasons."

48. L253-255: Is this known? This statement should be supported by appropriate references

We have rephrased the statement:

"Some efficient samples in this study were mainly from northwest China (see Sect. 3.2), and we cannot exclude the effect of feldspar content on ice nucleation activity when comparing dust particles from different deserts. Note that this is only a

possible conjecture based on very limited evidence, and more further studies are needed." (L268-271)

49. L258: "The near-surface…" This is unclear, here you compare ns, which is normalized to particle size/surface area, or am I misunderstanding you here?

In light of the difference in the vertical distributions of dust particles, we believe that there may be more large particles in the near-surface-collected samples than that collected in an aircraft. An explicit size dependence of surface ice active site density has been confirmed in this study. Hence, near-surface-collected samples may show higher $n_s(T)$ than the aircraft measurements.

The statement has been modified:

"…, the concentration of larger particles near the ground is higher (Maki et al., 2019), suggesting that the n_s (T) values of near-surface-collected samples may be higher, i.e., they may show more efficient INP activity than the aircraft measurements." (L272-274)

50. L260: "…to be more active INPs" compared to dust?

The statement is not clear and it has been rephrased:

"…which are considered to be more active INPs than dust particles…" (L275)

51. L262: Delete "green"

Deleted.

52. L264-265: Can be reduced to 2-3 main references, as you detail these studies in Sect. 3.4.

We followed the comment and reduced the number of references (L279).

53. L270: Replace "population" by "number concentration"

Replaced (L285).

54. L272: Add: "…ice nucleation"

Added (L287)

55. L274: Delete space in front of D50

Deleted (L289).

56. L275: "and" should not be italicized

We followed the comment and reset the font (L290).

57. L277: Replace "warm" by "high"

Replaced (L292).

58. L280: Do the indicated uncertainties correspond to standard deviations for the 12 samples? Please specify here and in the caption of Fig. 5.

We followed the comment and added a related definition and introduction in the revised manuscript and supplement information.

The indicated uncertainties correspond to the standard deviation of 12 samples at each temperature. Related information was added in L297 of the revised manuscript and in the caption of Table S3 of SI. There was no uncertainty information in Fig. 5. Thus, we didn't add more about it.

"The above uncertainties correspond to the standard deviation of 12 samples at each temperature." (L296-297)

59. L290-293: This contradicts your hypothesis presented in Sect. 3.3. that different feldspar content contributes to different freezing abilities." There is not sufficient evidence provided to claim/suggest a difference in mineralogical composition between the two transport pathways. I suggest to completely leave this out and focus on the aspect of the biological fraction, where direct measurements and support is provided by your data. Please see my main comment above.

We thank the referee for this kind suggestion and partly followed the comment.

In the revised version, we replotted the figure and added detailed comparison between two pathways, emphasizing the contribution of biological materials. However, the nucleation activity of all northwest samples was higher than that of the north samples, so that we think the possibility of mineral composition should not be completely ignored.

"However, it should be noted that the nucleation activity of all northwest samples was higher than that of the north samples, suggesting that there might be a difference in mineral composition (e.g., feldspar content), although it was far less important than the contribution of biological materials." (L311-313)

60. L296: Change "can't" to "cannot"

Changed (L316).

61. L304: Replace "Where" by "Here"

Replaced (L325).

62. L313: Replace "the first two lines" by "…between the lines of D50 = 5.6 μm and the ns curve for submicron particles."

Replaced:

"…between the fit lines of $D_{50}$= 5.6 μm and submicron particles." (L334)

63. L317: "1.0 ~ 3.2 μm" I assume this line corresponds to the average of the D50 = 3.2, 1.8 and 1.0 μm lines which overlap in Fig. 7a, right? This should be specified in the text. I also suggest to replace "~" by "-" and chose a color that is distinctively different to any color used for the individual lines to avoid confusion.

We followed the comments and rephrased the statement for clarity:

"… the $D_{50} = 3.2, 1.8 \ and \ 1.0$ μm lines were averaged into one, 1.0 - 3.2 μm-fit line, as shown in Fig. 7 (b)." (L335)

The "~" in "1.0 ~ 3.2 μm" has been replaced by "-" throughout the paper.

And the color of the "1.0 - 3.2 μm-fit line" is changed to dark green in Figure R1.2.

(Modified Fig. 7).

[Figure]

Figure R1.2. Modified Fig. 7

64. L319: "was less than 1 to 5 μm": Do you mean "below 5 μm"?

Rephrased.

"…less than 5 μm" (L338)

65. L322: contain more highly ice active minerals"

Sect. 3.5 was reorganized and this sentence was deleted.

66. L323: Delete "exactly"

Deleted.

67. L324: "-25 °C". From the figure it appears to be more likely "-29 °C".

We followed the referee's comment and corrected it.

"At temperatures below -29 °C, ..." (L342)

68. L324-325: "This phenomenon can…" If this was the case, why does your submicron parametrization deviate strongly from the quartz parametrization of Harrison et al. (2019) at higher temperatures? Is this because feldspar ice nucleation activity dominates at higher temperatures? This should be specified.

The ice nucleation activity at higher temperatures might be attributed to feldspar

and biological materials. However, we cannot tell how much these factors contributed based on available results. We pointed out existing uncertainties and interpreted the difference cautiously:

"At temperatures below -29 °C, the submicron-fit line coincides with the 12% quartz parameterization by Harrison et al. (2019), but is 1 to 2 orders of magnitude higher above this temperature." (L342-343)

69. L333: Replace "components" by "factors"

Replaced (L362).

70. L334-335: "…are more active…" Compared to what? The Atkinson et al. (2013) K-feldspar line?

Sect. 3.5 was thoroughly reorganized and this sentence was deleted.

71. L335: « Overall… » This statement seems misplaced and should be moved to Sect. 4

Corrected.

72. L342-343: Please see my comment above. Your data suggest that the difference is mainly driven by a difference in the biological material present on the dust particles from the two transport pathways.

We followed the comment and rephrased the statement in the revised version:

"… dust particles transported from China's northwest and northern deserts have different INP concentrations and ice nucleation efficiencies." (L374-375)

"And the average concentration proportion of heat-sensitive INPs was higher in northwest than in north pathway, indicating that the most discrepancy in nucleation activity between the two pathways was attributed to the abundance of heat-sensitive INPs, although the presence of different mineral fractions cannot be excluded." (L384-387)

73. L354: Replace "warm" by "high"

Replaced (L387).

74. L355: Are you trying to say that Asian dust has a higher abundance of biological material compared to desert dust?

We rephrased the sentence:

"These results not only explain the higher nucleation activity exhibited by our samples at relatively high temperatures (above -15 °C), but also emphasize the important role of biological materials during the seasonal Asian dust transport process." (L387-389)

75. L362: "…emphasizing the importance…" I suggest to tune this down a little bit: "…potentially suggesting the importance of larger particles for cloud formation."

Corrected. (L394)

76. L363: "as particle size reflects … » This statement should be support by references.

There are clearly different mineral types, contents and assemblages between different-sized fractions, thereby indicating clear grain-size dependence of mineral composition (Krippner et al., 2015; Xie et al., 2020).

However, these discussions are hardly to be found within the field of atmosphere and ice nucleation researches, so that we deleted this sentence.

References:

- Krippner, A., Meinhold, G., Morton, A. C., Russell, E., and von Eynatten, H.: Grain-size dependence of garnet composition revealed by provenance signatures of modern stream sediments from the western Hohe Tauern (Austria), Sedimentary Geology, 321, 25-38, 10.1016/j.sedgeo.2015.03.002, 2015.

- Xie, Y., Liu, L., Kang, C., and Chi, Y.: Sr-Nd isotopic characteristics of the Northeast Sandy Land, China and their implications for tracing sources of regional dust, Catena, 184, 10.1016/j.catena.2019.104303, 2020.

77. L364-365: "Due to the single requirement…" Unclear what you mean, please rephrase.

We rephrased the sentence:

"Since only particle size distribution is required as an input without particle mineralogical compositions, the new particle size-based parametrizations can be widely applied in models, …" (L396-498)

78. Fig. 3: Why are there two blue lines coming from the northwest pathway, i.e. lie on top of the red trajectories?

The blue lines represent the north pathway and consist of three samples (M3, M5, D6). These 72-hour back trajectories were initiated at the beginning of each sampling period, and started a new trajectory every 1 or 2 hours until the end of the sampling period. There were 12 lines in sample M5 (see R1.3), and the 2 (actually was 3) blue lines were part of them, indicating that the wind direction changed during the sampling period. However, most dust particles were originated from the north pathway.

[Figure]

Figure R1.3 Air mass trajectories of sample M5.

79. Fig. 5: Include space between value and unit, i.e. "10 °C"

Corrected.

---

## Author Comment (AC2) · 12 Jan 2021

*The authors are grateful to the editor and referees for their careful reading and constructive suggestions that substantially help to raise the quality of our manuscript. Below we address each of the comments listed in blue font. Our answer is listed in black font and revised text is listed in green font. The number of lines in our answers is based on the revised manuscript, and the amendments were marked with a highlight in the revised version.*

**Referee #2:**

In this manuscript, a study is introduced for which size segregated airborne aerosol samples, sampled during desert dust events in China, were examined concerning the particles ability to act as ice nucleating particle (INP). Besides for the effect of size, also possible contributions from biogenic materials were examined.

The study was well done and is well presented. The results are very timely. I have a few concerns, but nothing really major. Major revisions are only recommended (instead of minor revisions) due to the number of comments. However, after addressing my below comments, the manuscript will very likely be suited for publication in ACP.

We appreciate the referee's affirmation and comments on our work. The comments were responded point-by-point in the following contents, and the manuscript was completely revised. We believe the referee's concerns have been addressed.

**major remarks:**

1. line 23-24: What is the advantage of a size dependent parameterization over a single n_s curve that could be used with the total surface area of an aerosol? This could be discussed later in the text. Also, you could estimate which error would be done if one used such a single n_s curve, compared to your higher size resolved information, for which, however, more modeling capacity will be needed.

   We thank for the comment.

   A single $n_s(T)$ parameterization treats the freezing activities of all particles as homogeneous, ignoring the differences between different particle sizes. In contrast, size-resolved parameterizations classify particles into categories based on their size

dependent nucleation activity, allowing a better description of the properties of particles, which is particularly suitable for mineral dust.

We added a discussion as below:

"Compared with mineral composition-based parameterizations, the advantage of particle size-based curves is that we do not need to know the complex mineralogical composition of dust. Only the particle size distribution, a widely monitored parameter, is required in size-based prediction. Furthermore, there is a vertical distribution of mineral dust in the atmosphere (Maki et al., 2019), implying potential different contributions of these various size particles in cloud formation. Our size dependent parameterizations can provide more refined simulation and prediction in theory, which needs to be confirmed in further model studies." (L363-368)

2. section 2.1: You do discuss losses in a MOUDI a bit. But I still wonder if the filters put on the MOUDI-stages did not change the collected sizes, as this is sensitive to the distance between the plates. Also, the filters are much larger than the collection area of a MOUDI - were the filters cut or how was this issue dealt with? And how were they kept in place?

We thank for this comment.

It is really important to ensure that the jet-to-plate distance is the same as the design. According to the handbook, the MOUDI was designed for and calibrated with 0.001 inches (i.e., 0.0254 mm) thick substrate material. The polycarbonate filters (47 mm Nuclepore, Track-Etch Membrane, 0.2 μm pores, Whatman) are thin (~0.02 mm) so they do not affect the impactor jet-to-plate distances.

The impaction plates are 47 mm in diameter and the substrates are 47 mm as well. Thus, no need to cut the filters. As shown in Fig. R2.1, a polycarbonate filter is clamped into the holder by the hold-down ring. All these operations are carried out in strict accordance with the instrument manual.

[Figure]

Figure R2.1. Photograph of a collection plate fixed with a polycarbonate filter

3. line 149: It is clear that either diameters as measured by the SMPS needed to be adapted, or the aerodynamical diameters as measured by APS and as selected by MOUDI. But without checking literature thoroughly, I would have expected that geometrical diameter is the one most previous studies refered to, not aerodynamic diameter, when retrieving n_s. Please check this and give an estimation of the deviation this may cause.

We gratefully thank the referee for such important point and followed the comment. The particle diameter (i.e., particle surface area) is a main uncertainty source for the calculation of surface ice active site density, and the diameter should be specified when comparing the $n_s(T)$ values among different studies.

We searched the classical literatures on $n_s(T)$ calculation, including the papers by earlier definers and the papers commonly used to compare parameterizations. Several different measurement methods were used to determine the specific surface area in previous studies, such as particle mobility (Connolly et al., 2009; Niemand et al., 2012), gas adsorption (BET, Hiranuma et al., 2015; Atkinson et al., 2013), laser diffraction (Atkinson et al., 2013), and dynamic light scattering (DLS, Hiranuma et al., 2014), and optical probe (Price et al., 2018) techniques. Two main diameters, geometric and BET-inferred diameters (derived from BET-inferred surface area by some assumptions), were adopted in calculating $n_s(T)$, although some studies did not mention which particle sizes they used. It is clear that the BET-inferred surface area is typically larger than simplified spherical estimation, resulting in a lower $n_s(T)$ value if employed (Hiranuma et al., 2015).

On the one hand, some studies explicitly converted the particle size to geometric diameter. Niemand et al. (2012) converted both the aerodynamic and equivalent mobility diameter into a volume equivalent sphere diameter (i.e., geometric diameter, assuming that the bulk density is 2.6 g cm-3 and the dynamic shape factors are between 1.1 and 1.4). Hiranuma et al. (2014) converted the aerodynamic diameter to a volume equivalent diameter to calculate the geometric total surface area.

On the other hand, some studies used other particle sizes or ignored the differences between particle sizes. Connolly et al. (2009) measured the dust particle size distribution by SMPS, and did not talk about the conversion of stokes diameter when calculating the ice-active surface site density (IASSD). Hoose and Möhler (2012) summarized previous studies and neglected the deviations when the reported size is the mobility diameter instead of the geometric diameter in their calculation and comparison. Atkinson et al. (2013) used a laser diffraction technique to get the specific surface area, which was 3.5 times smaller than that determined by BET.

We evaluated the bias of the results calculated using aerodynamic and geometric diameter. The geometric diameter can be converted from its aerodynamic diameter as:

$$D_{\text{ae}} = D_g \sqrt{\frac{\rho_p C_{\text{g}}}{\rho_0 C_{\text{ae}} \chi}}$$

where $D_{\text{ae}}$ is aerodynamic diameter, $D_g$ is geometric diameter (i.e., the volume equivalent diameter), $\rho_0$ is unit density (1 g cm$^{-3}$), $\rho_p$ is the particle density, $\chi$ is the dynamic shape factor, $C_g$ and $C_{\text{ae}}$ are the Cunningham slip correction factors associated with the geometric and aerodynamic diameters, respectively.

Table R1.1. The deviation of calculations between the geometric and aerodynamic diameters

| $\rho_p$ (g cm$^{-3}$) | $\chi$ | $D_{\text{ae}}$ | $D_g$ | $n_s(T)^{\text{a}}$ |
|---|---|---|---|---|

| | | | | | |
|---|---|---|---|---|---|
| 2.6 | 1.1 | $D_{ae}$ | $D_g = 0.65\ D_{ae}$ | $n_{s,g} = 2.36\ n_{s,ae}$ | $n_{s,ae} = 0.42\ n_{s,g}$ |
| 2.0 | 1.1 | $D_{ae}$ | $D_g = 0.74\ D_{ae}$ | $n_{s,g} = 1.82\ n_{s,ae}$ | $n_{s,ae} = 0.55\ n_{s,g}$ |
| 1.8 | 1.1 | $D_{ae}$ | $D_g = 0.78\ D_{ae}$ | $n_{s,g} = 1.64\ n_{s,ae}$ | $n_{s,ae} = 0.61\ n_{s,g}$ |
| 1.5 | 1.1 | $D_{ae}$ | $D_g = 0.86\ D_{ae}$ | $n_{s,g} = 1.36\ n_{s,ae}$ | $n_{s,ae} = 0.74\ n_{s,g}$ |
| 2.6 | 1.4 | $D_{ae}$ | $D_g = 0.73\ D_{ae}$ | $n_{s,g} = 1.86\ n_{s,ae}$ | $n_{s,ae} = 0.54\ n_{s,g}$ |
| 2.0 | 1.4 | $D_{ae}$ | $D_g = 0.84\ D_{ae}$ | $n_{s,g} = 1.43\ n_{s,ae}$ | $n_{s,ae} = 0.70\ n_{s,g}$ |
| 1.8 | 1.4 | $D_{ae}$ | $D_g = 0.88\ D_{ae}$ | $n_{s,g} = 1.29\ n_{s,ae}$ | $n_{s,ae} = 0.78\ n_{s,g}$ |
| 1.5 | 1.4 | $D_{ae}$ | $D_g = 0.97\ D_{ae}$ | $n_{s,g} = 1.07\ n_{s,ae}$ | $n_{s,ae} = 0.93\ n_{s,g}$ |

[a] $n_{s,g}$ and $n_{s,ae}$ are the surface ice active site densities associated with the geometric and aerodynamic diameters, respectively.

Table R1.1 shows the results of calculations using different particle densities ($\rho\_p = 1.5 - 2.6$ g cm$^{-3}$) and dynamic shape factors ($\chi = 1.1 - 1.4$, Niemand et al., 2012) when the slip correction factor is not considered. At a given particle density and dynamic shape factor, $D_g$ is 0.65 to 0.97 times $D_{ae}$, and $n_{s,g}$ is 1.07 to 2.36 times $n_{s,ae}$. Therefore, our $n_s(T)$ derived from the aerodynamic diameter is 0.42 to 0.93 times the value of $n_s(T)$ determined by the converted geometric diameter.

We choose to use the aerodynamic diameter rather than the converted geometric diameter to derive the $n_s(T)$ values for three reasons. First, the conversion between aerodynamic and geometric diameters requires knowledge of particle density and shape factor. However, the above two parameters are associated with the chemical composition, diameter and morphology of particles, and cannot be measured directly. There is large uncertainty when using estimated fixed values. In fact, the Cunningham slip correction factor, which is often neglected in calculations, is also an important factor for particles smaller than 1 μm. Second, the determination of geometric diameter is influenced by the wavelength of the measuring instrument. Third, the airborne particles collected in our measurement were mixed particles rather than pure mineral dust, and the size distribution was mainly detected by APS. We think the uncertainty would be reduced to the greatest

extent when using the aerodynamic particle size in calculation.

In a word, we use the aerodynamic diameter in calculating $n_s(T)$, and note that the uncertainty should be borne in mind when comparing our data with other studies. A related statement is added in the revised manuscript and Supplementary Information:

"Note that aerodynamic diameter was used in the calculation of $n_s(T)$, which is 0.42 to 0.93 times the value of $n_s(T)$ determined by the converted geometric diameter (see the SI for more details)." (L133-134)

References:

- Atkinson, J. D., Murray, B. J., Woodhouse, M. T., Whale, T. F., Baustian, K. J., Carslaw, K. S., Dobbie, S., O'Sullivan, D., and Malkin, T. L.: The importance of feldspar for ice nucleation by mineral dust in mixed-phase clouds, Nature, 498, 355-358, 10.1038/nature12278, 2013.

- Connolly, P. J., Möhler, O., Field, P. R., Saathoff, H., Burgess, R., Choularton, T., and Gallagher, M.: Studies of heterogeneous freezing by three different desert dust samples, Atmos. Chem. Phys., 9, 2805-2824, 10.5194/acp-9-2805-2009, 2009.

- Hiranuma, N., Hoffmann, N., Kiselev, A., Dreyer, A., Zhang, K., Kulkarni, G., Koop, T., and Möhler, O.: Influence of surface morphology on the immersion mode ice nucleation efficiency of hematite particles, Atmos. Chem. Phys., 14, 2315-2324, 10.5194/acp-14-2315-2014, 2014.

- Hiranuma, N., Augustin-Bauditz, S., Bingemer, H., Budke, C., Curtius, J., Danielczok, A., Diehl, K., Dreischmeier, K., Ebert, M., Frank, F., Hoffmann, N., Kandler, K., Kiselev, A., Koop, T., Leisner, T., Mohler, O., Nillius, B., Peckhaus, A., Rose, D., Weinbruch, S., Wex, H., Boose, Y., DeMott, P. J., Hader, J. D., Hill, T. C. J., Kanji, Z. A., Kulkarni, G., Levin, E. J. T., McCluskey, C. S., Murakami, M., Murray, B. J., Niedermeier, D., Petters, M. D., O'Sullivan, D., Saito, A., Schill, G. P., Tajiri, T., Tolbert, M. A., Welti, A., Whale, T. F., Wright, T. P., and Yamashita, K.: A comprehensive laboratory study on the immersion freezing behavior of illite NX particles: a comparison of 17 ice nucleation measurement techniques, Atmospheric Chemistry and Physics, 15, 2489-2518, 10.5194/acp-15-2489-2015, 2015.

- Hoose, C., and Mohler, O.: Heterogeneous ice nucleation on atmospheric aerosols: a review of results from laboratory experiments, Atmospheric Chemistry and Physics, 12, 9817-9854, 10.5194/acp-12-9817-2012, 2012.

- Niemand, M., Möhler, O., Vogel, B., Vogel, H., Hoose, C., Connolly, P., Klein, H., Bingemer, H., DeMott, P., Skrotzki, J., and Leisner, T.: A Particle-Surface-Area-Based Parameterization of Immersion Freezing on Desert Dust Particles, Journal of the Atmospheric Sciences, 69, 3077-3092, 10.1175/jas-d-11-0249.1, 2012.

- Price, H. C., Baustian, K. J., McQuaid, J. B., Blyth, A., Bower, K. N., Choularton, T., Cotton, R. J., Cui, Z., Field, P. R., Gallagher, M., Hawker, R., Merrington, A., Miltenberger, A., Neely Iii, R. R., Parker, S. T., Rosenberg, P. D., Taylor, J. W., Trembath, J., Vergara-Temprado, J., Whale, T. F., Wilson, T. W., Young, G., and Murray, B. J.: Atmospheric Ice-Nucleating Particles in the Dusty Tropical Atlantic, Journal of Geophysical Research: Atmospheres, 123, 2175-2193, 10.1002/2017jd027560, 2018.

4. line 170 ff: The description of the definition of the dust events needs to be adjusted: from Table 1, it seems that it was based on CMA observations, only. Otherwise M5, D6 and D7 would also not be dust events.

We thank the referee for this comment.

The definition of dust events was based on the combined result of 4 factors: $PM_{10}$ mass concentration (larger than 200 μg m$^{-3}$ lasting more than 2 hours for dust events), the volume concentration of coarse mode particles (mean concentration higher than 75 μm$^3$ cm$^{-3}$ for dust events), phenomenological dust storm observations operated by China Meteorological Administration (CMA, being reported as the largescale dust events), and the concentration of aluminium (Al) element. Only some important information was given in Table 1, and more detailed determination criteria were discussed in Table R1.2.

The average PM10 mass concentrations of sample M5, D6 and D7 were not as high as that of other samples. A determination criterion of dust events is, however, that $PM_{10}$ mass concentration was larger than 200 μg m$^{-3}$ for more than 2 hours. In addition, aluminium (Al) is usually selected to be an indicator of mineral dust, and the concentration of Al is much higher in dust events than in non-dust event (M4). Therefore, we think sample M5, D6 and D7 would also be dust events.

We presented these detailed discussions in the Supplementary Information and

added a description in the revised manuscript:

"… (see Table S1 in the SI for more detailed determination criteria)." (L181)

Table R1.2. The criteria used to distinguish between dust and non-dust events for 14 sets of samples (Table S1).

The two weather conditions, i.e., dust and non-dust events, were defined based on PM10 mass concentration (PM10 Mass Conc.), the volume concentration of coarse mode particles (Vol. Conc.), phenomenological dust storm observations operated by China Meteorological Administration (Observations by CMA), and the concentration of aluminium (Al).

| Sample ID | $PM_{10}$ Mass Conc.[1] | Vol. Conc.[2] | Observations by CMA [3] | Concentration of Al element ($\mu g\ m^{-3}$) [4] | Weather condition |
|---|---|---|---|---|---|
| M1 | True | True | True | 5.65 | Dust |
| M2 | True | True | True | 1.68 | Dust |
| M3 | True | True | True | 0.72 | Dust |
| M4 | True | False | False | 0.04 | Non-dust [5] |
| M5 | True | False | True | 0.12 | Dust |
| M6 | True | True | True | 1.45 | Dust |
| M7 | True | True | True | 1.07 | Dust |
| M8 | True | True | True | 1.01 | Dust |
| D2 | True | True | True | 0.14 | Dust |
| D3 | True | True | True | 0.77 | Dust |
| D4 | True | True | True | 0.39 | Dust |
| D5 | True | True | True | 0.59 | Dust |
| D6 | False | False | True | 0.13 | Dust |
| D7 | True | False | True | 0.17 | Dust |

Note: The weather condition of each sample was defined by a combination of the above 4 factors.

[1] $PM_{10}$ mass concentration: 'True' was defined as $PM_{10}$ mass concentration larger than

200 μg m$^{-3}$ for more than 2 hours.

$^2$ The volume concentration of the coarse mode particles (> 1 μm): 'True' was defined for mean concentration higher than 75 μm$^3$ cm$^{-3}$. The threshold was developed based on the measurements of 2004-2006 in Beijing. Asian dust loading has declined in recent years. Thus, this threshold is not mandatory.

$^3$ Phenomenological dust storm observations: China Meteorological Administration (CMA) provides predictions and observations on dust storm events that occurred in China. The dust events in Beijing identified in this study have been reported as the largescale dust storm events.

$^4$ Aluminium (Al) is usually selected to be an indicator of mineral dust because it is one of the most abundant constant elements in deserts. Thus, the concentration of Al is considered as an important factor to define dust events.

$^5$ Sample M4 was collected from the end of a continuous dust storm (M1, M2, and M3), i.e., during the removal process after a dust storm. High wind speeds can blow up large particles from the roads and other surfaces in the city. In addition, the air mass of M4 passed through the Bohai Sea before arriving in Beijing (Fig. S1), possibly bringing large marine particles. Although the average concentration of PM$_{10}$ for sample M4 was higher than those of samples M5 and D6, the concentration of Al in sample M4 was much lower compared to sample M5 and D6. Therefore, we classify sample M4 as a non-dust event, since it's not dominated by mineral dust.

5. lines 175ff: Were results from M4 included in this study at all? If yes, how? If not, why mention it? And why are M1, M2, M3 and M4 described here explicitly, but not all the other samples? A sentence or two on the samples, even if they are only summarized in groups, would be helpful, so that for each sample its specifics are clearer (like the one found in line 214 - you can repeat some of that information there again, but also add it here).

We agree with the referee and added some content in the revised manuscript.

The results of M4 (non-dust event) were only shown in Table 1 and Fig. 2 (a) as a comparison with the dust events. The sample M4 triggered freezing at much lower

temperatures than that in the dust events (see Fig. 2 (a)), indicating that mineral dust particles are efficient INPs.

Only a non-dust sample (i.e., M4) was collected in our experiment, that is why we discussed it separately here. However, such a description does cause unnecessary confusion to the reader. We followed the comment, added an overview of the samples, and rephrase the description about M4:

"Seven and 6 dust samples were collected in 2018 (M1, M2, M3, M5, M6, M7 and M8) and in 2019 (D2, D3, D4, D5, D6 and D7), respectively. Sample M4 was sampled from the end of a continuous dust storm period (M1, M2 and M3), and its air mass passed through the Bohai Sea before arriving in Beijing (see Fig. S1 in the SI). Therefore, sample M4 was not dominated by mineral dust, and it was classified as a non-dust event." (L181-185)

6. line 178: The data shown in Fig. 1 in "pale yellow" - are they all for size segregated filters? If so, why is the set of colorfully depicted data well above these curves for the three larges particle sizes? If this colorfully depicted data-set is one with particularly high ice activity, mention this explicitly.

We thank the referee for this comment.

The data shown in Fig. 1 in "pale yellow" are all the results of size-resolved filters. And this colorfully depicted data-set is a set of high ice active sample, which is a subset of the collected "pale yellow" samples. We rephrased the sentences to clarify the statement:

"Results of all freezing curves containing size-resolved airborne dust particles are presented in pale yellow in Fig. 1. Each curve corresponds to one sampled filter collected in dust events." (L186-187)

"Frozen fraction curves from sample M1, a set of high ice active samples in the collected samples, are also shown in Fig. 1, and each color depicts a different size class ranging from 0.18 to 10.0 μm." (L189-191)

"The collected size-resolved filter measurements are marked in pale yellow (named "Collected samples"), … As a subset of the "Collected samples", a set of filters

(sample M1) ranging from 0.18 to 10.0 μm are marked with different colors to illustrate the frozen fraction curves for size-resolved particles." (L715, the caption of Fig. 1)

7. Line 183-185: Based on f_ice curves, it cannot be judged which size class is more ice active. It is theoretically possible that there are just so many more (or less) particles in one size class than in others, so that this number is the overwhelming influence on the overall measured ice activity at that impactor stage, causing high (or low) f_ice. I suggest that you at least mention this in an additional sentence.

We agree with the referee that it cannot be judged which size class is more ice active based on $f_{ice}(T)$ curves. But at least, it clearly shows that there are indeed differences for different size particles. As suggested, we added the discussion about the number concentration of particles when analyzing the freezing curve:

"On the whole, large particle samples froze at higher temperatures, while smaller size samples froze at lower temperatures, indicating different ice nucleating abilities. Note that the effect of particle number size distribution on the frozen fraction curves needs to be considered, because $f_{ice}(T)$ is sensitive to the number of particles at a given size class." (L192-194)

8. line 194-195: "For mineral dust particles, their freezing temperatures were similar." Do you mean that the dust curves were all similar? That is not true, and you should normalize to dust surface area first before you make comments on that, anyway. You even mention that in the next sentences. This sentence here needs to be deleted or revised.

We thank the referee for this comment.

Sorry for the misunderstanding caused by this sentence. What we want to express is that the initial freezing temperatures of the dust samples were similar (~ -6 °C), not that the freezing curves were similar. The sentence has been rephrased in the revised version:

"For mineral dust samples, their initial freezing temperatures were similar (-6 ±

1 °C).” (L203-204)

9. line 203: Fig. 2 (b): Is this based on one sample, or an average from all? Mention this! And again, it might make more sense to show this for size-normalized data, but number is fine, here, too.

We followed the comment and added related description of Fig. 2 (b).

The results in Fig. 2 (b) are based on the average from 13 dust dominated samples, and detailed information is given in Table S2 in the SI. This content has been added:

“The trend of $N_{INP}$ with temperature and particle size based on 13 mineral dust-dominated samples is depicted in Fig. 2 (b), where the size-resolved $N_{INP}$ ranged from $10^{-2}$ to $10^{1}$ L$^{-1}$ of standard air (see Table S2 in the SI for more detailed information).” (L212-213)

“Bars represent the average $N_{INP}$ of 13 dust-dominated samples.” (L725, the caption of Fig. 2)

The size-normalized data was shown in Fig. 7 (a), i.e., the surface ice active site densities for different size classes. Thus, we chose to show the size-resolved $N_{INP}$ in Fig. 2 (b).

10. line 214: Why did you choose to only show results for this subset of samples? This needs to be justified in the text, otherwise the impression could arise that you did “cherry picking”, i.e., only displaying those samples that fit what you want to see.

When discussing the difference of INPs between two transport pathways, 4 samples (M6, M7, M8 and D7) were thought to be from the northwest pathway and 3 samples (M3, M5 and D6) were classed as coming from the northern area. As shown in Fig. R2.2, the trajectories of the above 7 samples (solid red and blue lines) were relatively concentrated and can be well distinguished from each other. The transport trajectories of other 6 dust-dominated samples (M1, M2, D2, D3, D4 and D5, solid green lines) were mainly distributed in the middle between northwest and north pathways, and they were difficult to be classed into anyone pathways. To make the

results more concise and focus on the comparison of the two pathways, we only showed the above 7 samples, and the remaining samples were not given in Fig. 3. Now, Fig. R2.2 was added into the SI, and the following sentence was added in the text:

"Other six samples (M1, M2, D2, D3, D4 and D5) were not followed these two pathways (see Fig. S2 in the SI), and will not be discussed in this section." (L226-227)

[Figure]

Figure R2.2. (Fig. S2) Air mass trajectories for different samples originated from the northwest (M6, M7, M8 and D7; solid red lines), north (M3, M5 and D6; solid blue lines) and other (M1, M2, D2, D3, D4 and D5; solid green lines) transport pathways.

11. line 215: Is Sample D7 included in Fig. 3 (b)? To me, only 6 curves are visible in Fig. 3 (b), too. Please check, and if D7 is not included, mention this in the text.

Sample D7 is indeed included in Fig. 3 (b), this curve, however, is overlapped with the other three samples. We recolored the sample D7 to green in Fig. R2.3 (only the color of sample D7 is different from that in Fig. 3 (b)). It is clearly shown that the 4 northwest samples are very similar, and there are 7 curves in this figure.

[Figure]

Figure R2.3. INP concentrations compared for the northwest and north pathways (different from that in Fig. 3 (b), sample D7 is colored by green).

12. line 217: The airborne INP concentration depends on getting dust suspended in the air. So even the same source region can yield different airborne concentrations for different conditions such as changing wind speed. Mention this additional restriction. Different shapes, however, really depict different types of INP. (Like, based on what you describe later, a biogenic component in the "red" samples.)

We agree with the referee that the collected dust samples from the same source region might yield different INP concentrations due to varying atmospheric transport conditions.

We followed this constructive comment and added some discussion about it:

"…, indicating that there might be differences in the types of INPs. However, we need to keep in mind that due to the influence of factors such as changing wind speed during the transport process, the atmospheric concentration of dust from the same source region may be different, leading to a varying concentration of atmospheric INPs." (L230-233)

13. line 233: Concerning Fig. 4 (a) and (b), it seems as if here only 10 different data-sets are shown. Is that correct, and if yes, why were the other data not shown? This is in line with my comment concerning line 214.

We thank the referee for his/her careful reading.

Eleven dust datasets are shown in Fig. 4 (a) and (b), and the results for sample D5 and D7 are not given. As described in L224, the size distribution data of sample D7are partially missing so that surface area could not be fully derived. The surface area of particles in sample D5 might be overestimated, thus we decided not to use it in further calculation about $n_s(T)$. Therefore, 11 dust samples are included when discussing surface ice active site density of dust particles.

We have added this in the revised version:

"In the present study, the total $n_s(T)$ values (11 dust samples without sample D5 and D7) span…" (L250)

14. line 235: It would be more informative to mention the temperature span at a single temperature, as this overall span depends on the measurement method you use! (It varies with the amount of air you collect, the surface area (hence the size distribution) and, to a lesser extent, to the number of droplets you examine, but NOT on any characterization of the INP.)

We agree with the referee that it is more rigorous to describe the temperature range at a single temperature. It has been modified in the revised manuscript:

"… span 2 orders of magnitude from $10^5$ to $10^7$ m$^{-2}$ at -15 °C." (L250)

15. line 286: Again, why is only a subset of all 14 sets of filter samples shown?

Figure 6 is plotted to compare the effect of heat treatment on freezing properties of northwest and north samples, so that only 4 northwest and 3 north datasets are shown. Please see our responses to the tenth and eleventh major remarks.

16. line 292-293: "This may suggest that after heat-sensitive INPs was removed, the two transport pathways are now dominated by similar material, which is probably mineral dust." I totally agree - but that makes any discussion of different feldspar contents, which you did above, futile. Please check the content of the text for consistency! Or, when you mention feldspar, already point out that this may not be important as the importance of the biogenic content will be discussed below.

We followed the comment and rephrase the sentence:

"This may suggest that after heat-sensitive INPs were removed, … mineral dust. However, it should be noted that the nucleation activity of all northwest samples was higher than that of the north samples, suggesting that there might be a difference in mineral composition (e.g., feldspar content), although it was far less important than the contribution of biological materials." (L310-313)

17. line 299-300: What does that sentence refer to (in Sect. 3.3)? Please explain what you mean.

Three hypotheses are proposed in Sect. 3.3 to explain why samples in this study were more active for temperatures above −15°C, and the contribution of biological materials is an important factor. Sect. 3.4 confirmed the contribution of heat-sensitive INPs (mostly attributed to proteinaceous biological materials).

The sentence has been modified:

"This conclusion also confirms the hypotheses in Sect. 3.2 and 3.3, i.e., biological materials made a substantial contribution to ice nucleation activity above -15 °C." (L320-321)

18. line 223ff: My advice is to not overinterpret such observations. There are measurement uncertainties on all of these curves, and there are different approaches. The n_s derived in a lab-study from mineral dust samples refers to the surface area of dust particles, only, while in your study, you naturally have to refer to the surface area of the total aerosol. Also, you used the aerodynamic diameter as the reference, while this, to my understanding, was not the case in the other studies. So please be careful when discussing such details.

We agree with the referee and reorganized Sect. 3.5 completely as suggested.

Only the paragraph describing the uncertainties is given below. Please see the revised manuscript for complete modification.

"We note that the quantitative mineralogical composition was not investigated in this study, so that we cannot explain the discrepancy accurately in terms of mineral

composition. On the other hand, while relatively minor, measurement and calculation uncertainties should be borne in mind when comparing our parameterizations with other curves as well. First, different experimental methods introduce measurement errors. A cold stage-based technique was applied in this study, while cloud simulation chamber (Niemand et al., 2012), laminar flow tube (Niedermeier et al., 2015) and many other cold-stage instruments (with varying size/volume droplets; Atkinson et al., 2013;Harrison et al., 2019;Reicher et al., 2019) were used to measure the activated fractions of tested particles/droplets at a given temperature. Then, the investigated particles came from various sources and underwent different processing, including airborne-collected, surface-collected (sieved or milled) samples, and single mineral dust components. Next, the calculation of $n_s(T)$ depends on a key parameter, particle surface area, which refers to the surface area of dust particles in laboratory studies, while refers to the surface area of total aerosol particles in this study and in R19. Furthermore, we adopted aerodynamic diameter to obtain $n_s(T)$, which underestimated the result (0.42 to 0.93 times) compared with that determined by the converted geometric diameter." (L347-358)

19. line 333-335: How does that fit with the fact that you see such a high biogenic / proteinaceous fraction being responsible for the ice activity at higher temperatures??? Also: The deviation can be seen at high temperatures, where your fits are much above the mineral-dust parameterizations - mention that explicitly. Also: Fitting a straight line over such a broad T-range might be misleading.

We thank the referee for this comment.

Dust storms in East Asia are mainly concentrated in spring, when the plants germinate, grow and bloom. Our experimental results confirmed the biological contributions to ice nucleation activity of East Asian dust, especially at higher temperatures. Compared with reference single mineral, our airborne dust samples are more representative and can better reflect the INP characteristics of the actual atmosphere. The contribution of seasonal biological components should be

concerned, so that we chose to use the data containing biological materials rather than the after-heated dataset to fit the parameterizations.

For the straight-line fitting, first of all, this method is a reasonable simplification and has been widely adopted in previous studies (Niemand et al., 2012; Ullrich et al., 2017). Then, the goodness of fit index in our study is fairly good. We also agree with the referee that a more comprehensive and complicated fitting could be used in further studies.

20. line 340-341: As said above, this temperature range rather characterizes your method than saying anything about the INP. Give the span at a single temperature, as this signifies who different your different samples were.

We followed the comment and modified the similar statement about $n_s(T)$ throughout the manuscript.

"The total surface ice active site density, $n_s(T)$, spanned 2 orders of magnitude from $10^5$ to $10^7$ m$^{-2}$ at -15 °C." (L373)

21. line 346: "the common effect of the activity" - what do you mean by that. This could be elaborated somewhat more, maybe even in an additional sentence.

INP concentrations depend not only on the activity of particles in a specific size range, but also on the total number concentration of the same size particles. Therefore, the INP concentrations first increased rapidly with increasing particle size, and then levelled.

"Although larger particles had higher $n_s(T)$ , their atmospheric number concentration was much less than that of fine particles, i.e., INP concentrations depend on the combined effect of both individual particle' activity and the particle number at a given size." (L378-380)

22. line 350: Is that really what you find. You argued with different feldspar content at some point, then with different biogenic content, and now you summarize all this in saying "all desert dusts are the same". Make the message of your text consistent

throughout the manuscript.

Corrected.

"We also demonstrated that the differences of both the total and size-resolved $n_s(T)$ values of natural mineral dust particles from East Asia, North Africa, and Eastern Mediterranean are within 1 to 2 orders of magnitude, suggesting similarities in ice nucleation activity." (L380-382)

23. line 362ff: "Larger particles are more active INPs, as particle size reflects the mineral composition to a large extent" - This is not necessarily true. If larger particles have a higher n_s, then it's true, but in general larger particles are more ice active because they have a larger surface area. Formulate this with more care.
Modified.

"Larger particles are normally more active INPs, as large particles have a higher $n_s(T)$ or a larger surface area" (L395)

**Technical issues and minor remarks:**

1. line 29: "relatively high temperatures" - say more precisely what you mean by that.
We followed the comment and rephrased the sentence:

"Mineral dust particles can act as ice nucleating particles (INPs) that trigger heterogeneous ice nucleation at relatively high temperatures and low relative humidities by providing nucleation surfaces to efficiently lower the energy barrier of critical ice embryo formation" (L28-30)

2. line 37: What exactly do you mean by "mid-level clouds". The use of "mixed-phase clouds" (as in the next sentence) seems more appropriate here.
We followed the referee's comment and replaced the "mid-level clouds" by "mixed-phase clouds" (L37).

3. line 52: "supplement for feldspar" - it is not clear what you mean, here. That the

two always occur together? That would not be correct, as feldspars are weathered clay minerals, and quartz is not a clay mineral. Please check and reword.

When explaining the ice nucleation process of dust from the perspective of mineral components, quartz is considered to be less active than felspars, but it is also important to explain the freezing observed at lower temperatures.

The sentence has been modified:

"As a major component in mineral dust, quartz is important to explain the freezing observed at lower temperatures, although it is less active than feldspars" (L52-53)

4. line 91: "stages … were detected" - wrong wording, needs to be changed.

We followed the referee's comment and rephrased the sentence as the suggestion from referee #1:

"We used stages 1 to 8 of the MOUDI with cut-points ($D_{50}$) ranging from 10 to 0.18 μm in aerodynamic diameters at a flow rate of 30 L min$^{-1}$ in this study." (L92-93)

5. line 108: Are you aware of the fact that the use of ultrasonic waves may change the structure of proteins and therewith change their functionality? (see. e.g., DeLeo et al., 2016) At least mention this in your text, so that future readers know about this issue when they consider repeating what you did.

We thank the referee for this comment and added the caution in the revised version:

"Note that the ultrasonic waves may influence the properties and function of proteins and change their bioactivity (De Leo et al., 2017)." (L112-113)

6. line 115: change "will not be expected and" to "is not".

Changed (L118).

7. line 120: This is the first time that active sites are mentioned, so you may want to add a few words on explaining what you mean by that.

We followed the comment and added a short introduction on ice active sites:

"Ice active sites are the preferred locations for ice nucleation on an INP, and …"

(L123)

8.  line 138: "Gross" seems a bit misplaced here. I suggest to use a different word. Or, as you use "gross" more often, what you mean by that.

    We followed the comment and replaced "gross" by "total" in the revised manuscript (L144, 235, 248, 249, 256, 262, 373, 380 and 733).

9.  line 287: Change "originated" -> "originating".

    Changed (L303).

10. line 289: "For example, $N_{INP}$ near temperature at -10 °C." This is not a complete sentence - check and correct.

    Modified.

    "… in some of the cases (e.g., $N_{INP}$ at ~ -10 °C)." (L306)

11. Fig. 2: "b" is missing in the legend for Chen et al. (2018b). Also, change the color either for the Bi et al. (2019) datapoint, and/or make it an open symbol (maybe an open star?), as it is difficult to discriminate between these data and those from D2.

    We followed the comment and modified the figure as shown below:

[Figure]

Figure R2.4. Modified Fig. 2